# Stochastic Window Transformer for Image Restoration

**Jie Xiao, Xueyang Fu,**[*] **Feng Wu, Zheng-Jun Zha**
University of Science and Technology of China, Hefei, China
ustchbxj@mail.ustc.edu.cn, {xyfu,fengwu,zhazj}@ustc.edu.cn

## Abstract

Thanks to the powerful representation capabilities, transformers have made impressive progress in image restoration. However, existing transformers-based methods do not carefully consider the particularities of image restoration. In general, image restoration requires that an ideal approach should be translation-invariant to the degradation, i.e., the undesirable degradation should be removed irrespective of its position within the image. Furthermore, the local relationships also play a vital role, which should be faithfully exploited for recovering clean images. Nevertheless, most transformers either adopt local attention with the fixed local window strategy or global attention, which unfortunately breaks the translation invariance and causes huge loss of local relationships. To address these issues, we propose an elegant stochastic window strategy for transformers. Specifically, we first introduce the window partition with stochastic shift to replace the original fixed window partition for training. Then, we design a new layer expectation propagation algorithm to efficiently approximate the expectation of the induced stochastic transformer for testing. Our stochastic window transformer not only enjoys powerful representation but also maintains the desired property of translation invariance and locality. Experiments validate the stochastic window strategy consistently improves performance on various image restoration tasks (deraining, denoising and deblurring) by significant margins. The code is available at https://github.com/jiexiaou/Stoformer.

## 1 Introduction

Image restoration aims to recover latent clean images from their noise-polluted counterparts, which lays the foundation for various vision tasks. Generally, image restoration methods should satisfy a constraint: *an ideal approach should remove undesirable degradation irrespective of its position within the image.* In other words, maintaining the translation invariance is the fundamental requirement for image restoration, which makes convolutional neural networks (CNNs) well-suited for this specific vision task [56, 23, 9, 73, 70, 61, 66, 58, 72, 6, 15]. In comparison with multi-layer perceptrons (MLPs), the distinct characteristics of CNNs [26, 22, 17] are their built-in locality and weight sharing, resulting in the desirable property of translation invariance. These two priors play a vital role in image restoration, i.e., pixels within local regions tend to exhibit strong correlations while translation invariance is one of the desiderata for the potential image restoration method.

On the other hand, transformer-based image restoration methods [2, 54, 33, 63] are not compatible with the translation invariance and locality. The patterns that transformers adopted for recovering high-resolution images can be summarized as small patch with global attention (e.g., $48 \times 48$ for IPT [2]) and large patch with local attention (e.g., Uformer [54] and SwinIR [33]). Nevertheless, neither of these two patterns can meet the requirements of translation invariance or locality. Here we detail

---

[*]Corresponding author.

36th Conference on Neural Information Processing Systems (NeurIPS 2022).

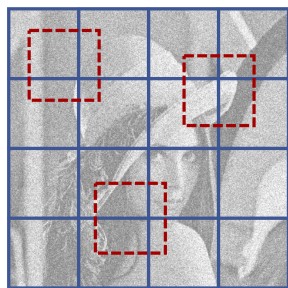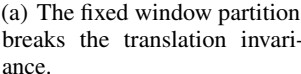

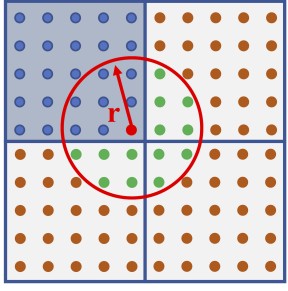

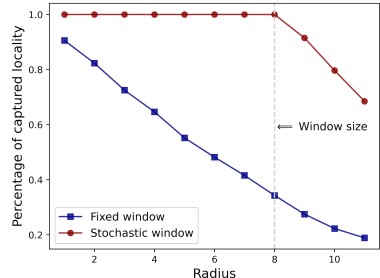

(a) The fixed window partition breaks the translation invariance.

(b) The fixed local window cannot faithfully capture local relationships.

(c) The comparison of percentage of captured locality between the fixed and stochastic window strategy.

Figure 1: Illustration of broken translation invariance and lost locality of the fixed window strategy.

the latter pattern while the former one is included in the supplemental material. The latter pattern partitions the feature map into fixed non-overlapped windows and self-attention is restricted inside these local windows. However, we argue that the fixed partition strategy will break the translation invariance and further lead to huge loss of local relationships. As shown in Figure 1(a), for the fixed window partition strategy, translation invariance can only be ensured among specific partitioned windows (blue windows)[2] rather than general windows with arbitrary shifts (red windows), which disagrees with the translation invariance. The reason behind this is that the fixed window partition imposes the artificial prior to windows with specific shifts, which turns out to break the translation invariance. Figure 1(b) intuitively illustrates that the fixed window partition cannot capture intact local relationships. Without loss of generality, we consider the Euclidean distance, based on which the neighbor space will be the region encompassed by a circle, in 2D plane to measure locality. For a certain token (central red dot), many tokens (green dots) within its neighbor space (encompassed by the red circle with radius $r$) cannot participate in the calculation of self-attention, which leads to tremendous loss of local relationships. As the neighbor space expands, i.e., increasing radius $r$, the loss will be more severe. Figure 1(c) quantitatively depicts the proportion of local relationships that are captured by the fixed window partition over the total relationships existing in the neighbor space. Ideally, when the feature map is divided into non-overlapping windows of size $s \times s$, it is expected that the relationships of token pairs with distance less than $s$ can be faithfully captured. However, we can observe that there exists significant loss of local relationships even within the neighbor space whose radius is much smaller than $s$. A seemingly straightforward solution to this problem is to adopt the sliding window strategy to scan the whole feature map using the local window like CNNs. However, this strategy incurs huge overhead in terms of both memory and computing speed.

In this work, we propose a novel stochastic window strategy to impart translation invariance to the transformer and make it faithfully model local relationships. Specifically, instead of partitioning the feature map into fixed non-overlapped windows, we choose to cover the whole feature map with a stochastically shifted window partition (see Figure 2). By introducing stochastic shifts, all windows are treated equally so that the translation invariance and locality can be ensured. As shown in Figure 1(c), our stochastic window strategy is able to capture complete local relationships until the radius of neighbor space reaches the window size. It is noteworthy that unlike the sliding window strategy, the time and memory overhead consumed for training stochastic window transformer is comparable to the fixed window strategy. For testing, we further design the layer expectation propagation algorithm to approximately marginalize the stochastic shifts, during which the translation invariance and locality can also be ensured.

In conclusion, the contributions are threefold:

- We analyze the phenomena of translation invariance breaking and local relationships loss in existing transformer-based image restoration approaches. To the best of our knowledge, this is the first work to point out the importance of translation invariance for image restoration.

---

[2]These windows are processed using shared weights, e.g., $W_Q, W_K, W_V$ for $Q, K, V$.

- We propose a new stochastic window strategy, which comprises stochastic windows for training and layer expectation propagation for testing, to compensate for the broken translation invariance of transformers and enable them to faithfully model local relationships.

- Extensive experiments on various tasks, e.g., deraining, denoising and deblurrring, validate that the restoration performance can be consistently improved by equipping our stochastic window strategy.

## 2 Related works

**Image Restoration.** Image Restoration [67, 11, 5, 36, 34, 38, 62, 40, 29, 28] aims to restore the clean image from its degraded version. In recent years, remarkable progress against traditional model-driven methods [4, 53, 16] has been achieved due to the development of deep learning technologies [27], especially CNNs [26, 22, 17]. Instead of relying on preset image priors, learning based methods directly learn to project from noisy to clean ones from a large collection of noisy-clean image pairs. Numerous representative CNNs have sprung up across various classical image restoration tasks, including image denoising [25, 60, 68, 66, 7, 74, 32], image super-resolution [72, 6, 15, 20, 75], image deblurring [9, 23, 39, 23, 47, 65, 43, 42], image deraining [58, 50, 13, 64, 51, 41, 30, 14], etc. In general, the translation invariance (derived from weight sharing) and locality have been hard-coded into the inherent structure of CNNs so that it seems CNNs are well-suited for image restoration tasks. However, compared with Transformer, CNNs are restricted by their limited flexibility.

**Vision Transformer.** Recently, with great success of transformer [49, 8] in the NLP field, Vision Transformers [10, 48, 52, 35, 57, 59, 3] have also been prevalent in vision community. ViT [10] treated image patches as token sequence and applied the vanilla transformer on it for image classification. With the goal of bringing in reasonable priors into transformer to improve efficiency, Swin [35] introduced local window based attention and established a hierarchical architecture. Inspired by the key insights from high-level vision, a few transformers [2, 33, 54, 63, 55] for low-level vision have arose. But most of them directly transfer high-level designs (e.g., local attention) into low-level vision without careful consideration of its particularities. As we discuss above, unlike CNNs, transformers do not possess the translation invariance and intact locality. In general, low-level tasks usually require to accomplish pixel-level regression, where more strict locality and translation invariance are expected. Besides, the CNNs-style sliding window strategy will incur huge burden in terms of memory and computing speed. Therefore, an efficient mechanism to remedy the broken translation invariance and locality of transformer is essential for low-level vision.

## 3 Stochastic Window Transformer for Image Restoration

Transformers have attained impressive performance for image restoration due to their strong representation ability. Given the quadratic complexity in both computation and memory usage, transformers for image restoration tend to employ the local attention. Except for high-efficiency, local attention should have been designed to model local relationships. However, as we analyzed before, local attention breaks the ideal translation invariance and further leads to huge loss of local relationships. As shown in Figure 2, for a feature map, the window partition with arbitrary shift $(\xi_h, \xi_w)$ contains a comparable quantity of local relationships and implies the translation invariance among divided windows. In other words, all the window partitions are equally informative in terms of translation invariance and locality and should be treated equally. Therefore, it is unreasonable for the fixed window strategy to express infinite favoritism towards the certain partition (e.g., $(0, 0)$ or $(\frac{s}{2}, \frac{s}{2})$) while simply discarding others. In this work, we propose the stochastic window strategy, by which all the window partitions are treated fairly. The network can be trained based on local attention but with stochastic rather than fixed shift. Therefore, transformer with the stochastically shifted window can be trained as efficiently as the fixed window. At test time, we propose the layer expectation propagation algorithm to approximate the expectation of the introduced stochastic shift, which also helps to maintain the desired property of translation invariance and locality.

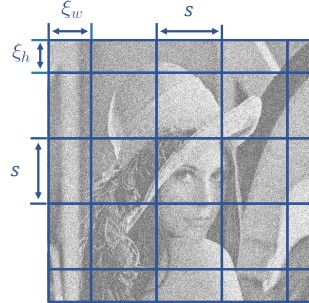

Figure 2: Window partition with shift $(\xi_h, \xi_w)$. $s$ is the spatial size of local window.

## 3.1 Stochastic Window Strategy

Stochastic window strategy aims to impart the translation invariance to transformer and make it faithfully exploit local relationships. Canonical transformer consists of alternate layers of self-attention (SA) and MLP. In order to promote efficiency, transformer tends to employ local window based attention and shifted window strategy is utilized to allow inter-window connections. Specifically, the feature map is partitioned into non-overlapped windows and then SA is computed within local windows. Suppose the window partition is denoted by $\mathrm{Par}(\cdot; s, \xi_h, \xi_w)$, where size of the local window is $s$ and shift of the whole window partition is $(\xi_h, \xi_w)$ (see Figure 2). With these notations, the shifted window transformer can be reformulated as:

$$
\begin{aligned}
z_l &= \mathrm{SA}(\mathrm{Par}(x_{l-1}; s, 0, 0)) + x_{l-1}, \\
x_l &= \mathrm{MLP}(z_l) + z_l, \\
z_{l+1} &= \mathrm{SA}(\mathrm{Par}(x_l; s, \frac{s}{2}, \frac{s}{2})) + x_l, \\
x_{l+1} &= \mathrm{MLP}(z_{l+1}) + z_{l+1},
\end{aligned}
\tag{1}
$$

where $z_l$ and $x_l$ denote the output feature of SA and MLP for layer $l$, respectively. To keep notation uncluttered, we overlook layer normalization in Equation (1). We figure out that all the window partitions are equally informative and preserving only one configuration each layer will lead to the broken translation invariance and tremendous loss of locality. The thought of informative equality among all the window partitions leads us to the stochastic window transformer, which is computed as:

$$
\begin{aligned}
z_l &= \mathrm{SA}(\mathrm{Par}(x_{l-1}; s, \xi_h^l, \xi_w^l)) + x_{l-1}, \quad (\xi_h^l, \xi_w^l) \sim \mathbb{U}(\Re_s), \\
x_l &= \mathrm{MLP}(z_l) + z_l,
\end{aligned}
\tag{2}
$$

where the set $\Re_s$ contains all possible shifts and $\mathbb{U}(\cdot)$ denotes the uniform distribution. Given the periodicity of the window partition, $\Re_s$ can be simplified as

$$
\Re_s := [0, \dots, s-1] \times [0, \dots, s-1],
\tag{3}
$$

where $\times$ means the Cartesian product. During training, $(\xi_h^l, \xi_w^l)$ are treated as i.i.d. random variables and sampled from the uniform distribution $\mathbb{U}(\Re_s)$. Suppose the total number of SA layer is $N$, stochastic shifts from different layers $\{(\xi_h^l, \xi_w^l)\}_{l=0}^{N-1}$ are deliberately designed to be independent so that faithful locality and translation invariance can be ensured on layer level. By this treatment, despite all possible shifts are taken into account for each individual layer, single forward propagation only requires a set of *sampled* shifts $\{(\xi_h^l, \xi_w^l)\}_{l=0}^{N-1}$ so that the stochastic window transformer can be trained efficiently. For testing, from Bayesian perspective, stochastic shifts should be averaged according to their posterior distribution. Hence, the exact inference procedure is:

$$
\begin{aligned}
F(x)^{\text{test}} &= \sum_{\xi_h^0, \xi_w^0, \dots, \xi_h^{N-1}, \xi_w^{N-1}} F(x; \xi_h^0, \xi_w^0, \dots, \xi_h^{N-1}, \xi_w^{N-1}) \mathbb{U}(\xi_h^0, \xi_w^0, \dots, \xi_h^{N-1}, \xi_w^{N-1}) \\
&= \sum_{(\xi_h^{N-1}, \xi_w^{N-1})} \cdots \sum_{(\xi_h^0, \xi_w^0)} F(x; \xi_h^0, \xi_w^0, \dots, \xi_h^{N-1}, \xi_w^{N-1}) \mathbb{U}(\xi_h^0, \xi_w^0) \cdots \mathbb{U}(\xi_h^{N-1}, \xi_w^{N-1}),
\end{aligned}
\tag{4}
$$

where $F(\cdot)$ denotes the function of the overall transformer and $\mathbb{U}$ is the uniform distribution. The derivation of Equation (4) follows $\{(\xi_h^l, \xi_w^l)\}_{l=0}^{N-1}$ are independent. Apparently, according to Equation (4), the single exact inference requires forward propagation exponential times so that the computational overhead of exact inference grows exponentially with the depth $N$, which is prohibitively expensive. Consequently, we turn to seek an approximate inference, which should also guarantee the translation invariance and intact locality, to replace the costly exact inference. Inspired by the approximation in Dropout [46], we propose the general layer expectation propagation algorithm to approach the original exact inference process. The layer expectation propagation is formulated as:

$$
\begin{aligned}
z_l^{\text{test}} &= \sum_{(\xi_h^l, \xi_w^l)} \mathrm{SA}(\mathrm{Par}(x_{l-1}^{\text{test}}; s, \xi_h^l, \xi_w^l)) \mathbb{U}(\xi_h^l, \xi_w^l) + x_{l-1}^{\text{test}} \\
&= \mathbb{E}_{(\xi_h^l, \xi_w^l) \sim \mathbb{U}} \left[ \mathrm{SA}(\mathrm{Par}(x_{l-1}^{\text{test}}; s, \xi_h^l, \xi_w^l)) \right] + x_{l-1}^{\text{test}}, \\
x_l^{\text{test}} &= \mathrm{MLP}(z_l^{\text{test}}) + z_l^{\text{test}}.
\end{aligned}
\tag{5}
$$

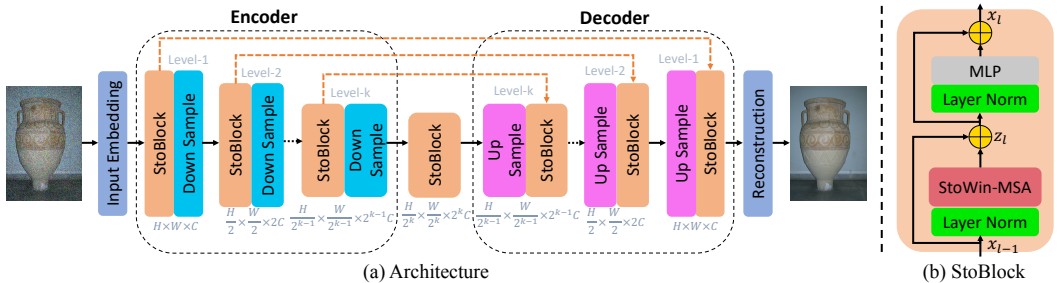

Figure 3: (a) The overall architecture of Stoformer††; (b) the structure of Stoformer†† block. StoWin-MSA is multi-head self-attention with stochastic window partition.

According to Equation (5), the single approximate inference merely requires one forward propagation of the expected signal, which accelerates inference considerably. We then examine whether the stochastic window strategy is able to keep the translation invariance and model intact local relationships.

**Translation Invariance.** During training, the stochastic window strategy treats arbitrarily shifted window equally and eliminates the particularity of certain windows introduced by the fixed window partition (blue windows in Figure 1(b)). Therefore, the stochastic window strategy can maintain the translation invariance during training. For testing, by the layer expectation propagation, each individual token is able to aggregate information from its neighbor space with shared weights like CNNs. Consequently, the translation invariance can also be well maintained. We also provide experimental evidence to support the translation invariance in Section 4.2.

**Locality.** We have argued that the fixed window partition will lead to huge loss of local relationships since the majority of equally informative window partitions are simply discarded. As shown in Figure 1(b), there exists some token pairs whose distance is small enough but they are not present in the same local window, which results in tremendous loss of locality. In contrast, with the stochastic window strategy, the window partition is randomly shifted, which will guarantee that any token pair will attend in the same window as long as their distance is smaller than the window size $s$. As shown in Figure 1(c), the stochastic window strategy can capture intact local relationships when the radius of neighbor space is not larger than window size while the fixed window partition suffers from severe loss of local relationships. For testing, the proposed layer expectation propagation also considers all the shifted window partitions so that intact locality can be ensured as well.

**Implicit Model Ensemble.** Compared with the typical fixed window strategy (in particular, the shifted window strategy), we can observe significant performance improvements from extensive experiments. In addition to the explanation from the compensation of the lost local information and the broken translation invariance, another explanation for performance improvements is that training with the stochastic window can be seen as training an ensemble of the fixed window transformer implicitly [18]. Each self-attention layer contains $w^2$ window partition, which results in $w^{2N}$ possible network combinations. For each training mini-batch, one of the $w^{2N}$ networks is sampled and then updated. For testing phase, all the networks are implicitly integrated using the proposed layer expectation propagation.

## 3.2 Network Architecture

The proposed stochastic window transformer provides an effective and efficient way to integrate the well-acknowledged priors, i.e., locality and translation invariance, into transformer. For image restoration tasks which require pixel-level regression, faithfully exploiting these priors is especially important. To validate the effectiveness of the proposed stochastic window strategy, we integrate it with the widely-used U-shaped architecture [44, 21]. The resulting transformer, named Stoformer††, not only enjoys the intrinsically strong representation ability of Transformer but also maintains the ideal translation invariance and intact local relationships. Figure 3 illustrates the overall architecture.

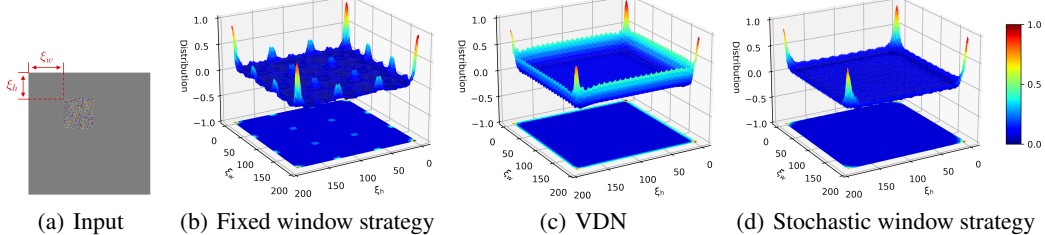

| (a) Input | (b) Fixed window strategy | (c) VDN | (d) Stochastic window strategy |

Figure 4: Toy Experiment. The stochastic window strategy (d) helps to remedy the broken translation invariance caused by the fixed window partition (b).

### 3.3 Implementation Details

**Loss Function.** The loss function adopted for training is the Charbonnier loss [1], whose mathematical expression is:

$$L(I', I) = \sqrt{||I' - I||^2 + \epsilon^2},$$ (6)

where $I'$ and $I$ are the restored and ground-truth image respectively. The constant $\epsilon$ is empirically set to $10^{-3}$.

**Training Detail.** Stoformer†† employs a four-level encoder-decoder structure. The numbers of StoBlock are $\{1, 2, 8, 8\}$ for level-1 to level-4 of Encoder and the blocks for Decoder are mirrored. The number of channel is set to 32 and the window size is $8 \times 8$. We train the network with Adam optimizer ($\beta_1 = 0.9, \beta_2 = 0.999$) with the initial learning rate $3 \times 10^{-4}$ gradually reduced to $1 \times 10^{-6}$ with the cosine annealing. The training samples are augmented by the horizontal flipping and rotation of $90°$, $180°$, or $270°$. Please refer to the supplemental material for task-specific settings.

## 4 Experiments

### 4.1 Experimental Setup

In this section, we validate the effectiveness of the stochastic window strategy. Except the previous state-of-the-art methods are included for comparisons, we also elaborate four Stoformer variants: Stoformer◇◇, Stoformer†◇, Stoformer◇†, Stoformer††. The first symbol aims to indicate whether the stochastic window is adopted for training(†: the stochastic window; ◇: the fixed window) and the second symbol represents whether the layer expectation propagation is adopted for testing(†: layer expectation propagation; ◇: the fixed window). In all experiments, we use the shifted window to specify the fixed window (Equation (1)). These variants are identical except for the aforementioned window strategy. In particular, Stoformer◇◇ is reduced to the traditional fixed window transformer and Stoformer†† is the proposed stochastic window transformer.

### 4.2 Toy Experiment

To illustrate the broken translation invariance of the fixed window strategy and the remedy of our proposed stochastic window strategy to this effect, a toy experiment is performed: we add a Gaussian noise patch with $\sigma = 50$ whose spatial size is $64 \times 64$ to an 8-bit pure color image with spatial size $256 \times 256$, the pixel value of which is constantly set to the medium value 127. We move the Gaussian noise patch with stride 1 and obtain total $192 \times 192 = 36864$ noisy images. Then we denoise these noisy images using the fixed and stochastic window transformer, respectively and the PSNR distributions with respect to $(\xi_h, \xi_w)$ are normalized to the fixed range $[0, 1]$ and plotted in Figure 4. To further highlight the translation invariance of stochastic window transformer, we also include a typical CNNs-based denoiser VDN [60], which can ensure the translation invariance well due to inherent structure of CNNs, as the inference. Figure 4 indicates clearly that there are great variations, especially for the interior, in the PSNR distribution of fixed window transformer while stochastic window transformer flattens the distribution significantly. Both stochastic window transformer and VDN contain the relatively uniform distribution for the interior, which indicates that the stochastic window strategy helps to maintain the desired translation invariance quite well.

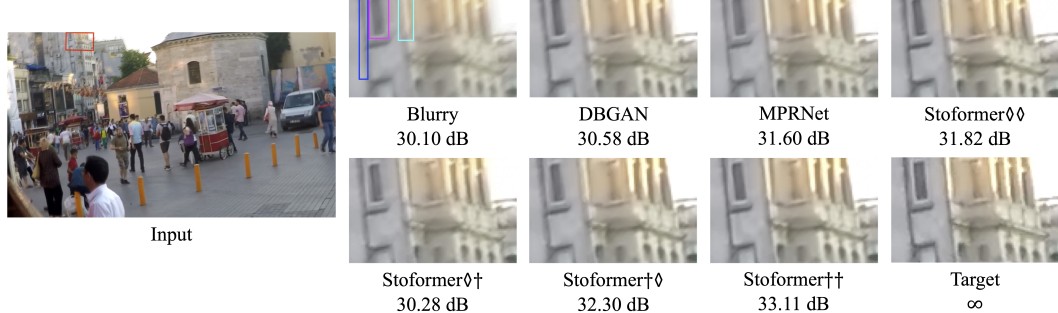

Figure 5: Visual comparison of image deblurring on the GoPro.

It is worth noting that all these distributions have variations on the image boundary, which can be attributed to the difference of the contexts of the image boundary.

### 4.3 Experiments on Image Deraining

Table 1: Quantitative results of image deraining on the SPA-Data.

| Method | SPA-Data | |
| --- | --- | --- |
| | PSNR | SSIM |
| GMM [31] | 34.30 | 0.9428 |
| DDN [13] | 36.97 | 0.9604 |
| SPANet [51] | 40.24 | 0.9811 |
| JORDER-E [58] | 40.78 | 0.9811 |
| RCDNet [50] | 41.47 | 0.9834 |
| SPAR [40] | 44.10 | 0.9872 |
| Uformer [54] | 47.84 | 0.9925 |
| Stoformer◊◊ | 47.80 | 0.9925 |
| Stoformer◊† | 46.95 | 0.9917 |
| Stoformer†◊ | 48.85 | 0.9935 |
| Stoformer†† | 48.97 | 0.9938 |

Table 2: Quantitative results of image deblurring. Stoformers are trained only on GoPro dataset.

| Method | GoPro | | HIDE | |
| --- | --- | --- | --- | --- |
| | PSNR | SSIM | PSNR | SSIM |
| Nah *et al.* [39] | 29.08 | 0.914 | 25.73 | 0.874 |
| DeblurGAN [23] | 28.70 | 0.858 | 24.51 | 0.871 |
| DeblurGAN-v2 [24] | 29.55 | 0.934 | 26.61 | 0.875 |
| DBGAN [69] | 31.10 | 0.942 | 28.94 | 0.915 |
| IPT [2] | 32.52 | - | - | - |
| MPRNet [62] | 32.66 | 0.959 | 30.96 | 0.939 |
| SPAIR [40] | 32.06 | 0.953 | 30.29 | 0.931 |
| Stoformer◊◊ | 32.80 | 0.959 | 30.73 | 0.937 |
| Stoformer◊† | 31.62 | 0.950 | 29.94 | 0.928 |
| Stoformer†◊ | 33.17 | 0.963 | 30.91 | 0.940 |
| Stoformer†† | 33.24 | 0.964 | 30.99 | 0.941 |

We validate the stochastic window strategy on image deraining task. Specifically, except the elaborate four Stoformer variants, existing seven deraining methods are included: GMM [31], DDN [13], SPANet [51], JORDER-E [58], RCDNet [50], SPAIR [40], Uformer [54]. All these methods are evaluated on SPA-Data [51] and the performance comparison is shown in Table 1. Due to the limited space, visualization of image deraining is included in the supplement material.

### 4.4 Experiments on Image Denoising

We conduct denoising experiments on the additive white Gaussian Noise benchmark datasets, which include Set12 [66], BSD68 [37], Urban100 [19], Kodak24 [12] and McMaster [71]. Following previous works [66, 60, 67], a single model is trained to tackle with various noise levels. Tables 3 and 4 report PSNR scores of existing methods as well as Stoformer variants for color and grayscale image denoising, respectively. The supplemental material contains visualization of image denoising.

### 4.5 Experiments on Image Deblurring

We also perform deblurring experiments on the benchmark datasets (GoPro [39] and HIDE [45]). The model is trained only on the GoPro dataset and directly evaluated on the HIDE. Table 2 presents PSNR and SSIM scores of different deblurring methods and Stoformers. Figure 5 presents an image deblurring example from GoPro [39]. More visual results are provided in the supplemental material.

**Remark.** From above experiments on various image restoration tasks, we can make the following observations and analyses:

Table 3: Gaussian color image denoising. A single model is learned for various noise levels.

| Method | CBSD68 | | | Kodak24 | | | McMaster | | | Urban100 | | |
|---|---|---|---|---|---|---|---|---|---|---|---|---|
| | $\sigma=15$ | $\sigma=25$ | $\sigma=50$ | $\sigma=15$ | $\sigma=25$ | $\sigma=50$ | $\sigma=15$ | $\sigma=25$ | $\sigma=50$ | $\sigma=15$ | $\sigma=25$ | $\sigma=50$ |
| IRCNN [67] | 33.86 | 31.16 | 27.86 | 34.69 | 32.18 | 28.93 | 34.58 | 32.18 | 28.91 | 33.78 | 31.20 | 27.70 |
| FFDNet [68] | 33.87 | 31.21 | 27.96 | 34.63 | 32.13 | 28.98 | 34.66 | 32.35 | 29.18 | 33.83 | 31.40 | 28.05 |
| DnCNN [66] | 33.90 | 31.24 | 27.95 | 34.60 | 32.14 | 28.95 | 33.45 | 31.52 | 28.62 | 32.98 | 30.81 | 27.59 |
| VDN [60] | 33.90 | 31.35 | 28.19 | - | - | - | - | - | - | - | - | - |
| FuncNet [36] | 34.28 | - | - | 35.25 | - | - | - | - | - | - | - | - |
| DRUNet [70] | 34.30 | 31.69 | 28.51 | 35.31 | 32.89 | 29.86 | 35.40 | 33.14 | 30.08 | 34.81 | 32.60 | 29.61 |
| Restormer [63] | 34.39 | 31.78 | 28.59 | 35.44 | 33.02 | 30.00 | 35.55 | 33.31 | 30.29 | 35.06 | 32.91 | 30.02 |
| Stoformer◇◇ | 34.34 | 31.73 | 28.52 | 35.32 | 32.91 | 29.83 | 35.53 | 33.35 | 30.34 | 35.04 | 32.83 | 29.66 |
| Stoformer◇† | 34.30 | 31.73 | 28.50 | 35.22 | 32.90 | 29.80 | 35.40 | 33.22 | 30.18 | 35.00 | 32.78 | 29.61 |
| Stoformer†◇ | 35.10 | 32.40 | 29.13 | 35.50 | 33.08 | 30.00 | 36.00 | 33.83 | 30.80 | 35.37 | 33.14 | 30.00 |
| Stoformer†† | 35.13 | 32.47 | 29.16 | 35.53 | 33.12 | 30.03 | 36.03 | 33.86 | 30.84 | 35.42 | 33.19 | 30.06 |

Table 4: Gaussian grayscale image denoising. A single model is learned for various noise levels.

| Method | BSD68 | | | Urban100 | | | Set12 | | |
|---|---|---|---|---|---|---|---|---|---|
| | $\sigma=15$ | $\sigma=25$ | $\sigma=50$ | $\sigma=15$ | $\sigma=25$ | $\sigma=50$ | $\sigma=15$ | $\sigma=25$ | $\sigma=50$ |
| DnCNN [66] | 31.62 | 29.16 | 26.23 | 32.28 | 29.80 | 26.35 | 32.67 | 30.35 | 27.18 |
| FFDNet [68] | 31.63 | 29.19 | 26.29 | 32.40 | 29.90 | 26.50 | 32.75 | 30.43 | 27.32 |
| IRCNN [67] | 31.63 | 29.15 | 26.19 | 32.46 | 29.80 | 26.22 | 32.76 | 30.37 | 27.12 |
| DRUNet [70] | 31.91 | 29.48 | 26.59 | 33.44 | 31.11 | 27.96 | 33.25 | 30.94 | 27.90 |
| Restormer [63] | 31.95 | 29.51 | 26.62 | 33.67 | 31.39 | 28.33 | 33.35 | 31.04 | 28.01 |
| Stoformer◇◇ | 31.94 | 29.51 | 26.62 | 33.60 | 31.38 | 27.90 | 33.26 | 31.02 | 27.96 |
| Stoformer◇† | 31.89 | 29.46 | 26.59 | 33.48 | 31.09 | 27.85 | 33.18 | 30.96 | 27.92 |
| Stoformer†◇ | 32.57 | 30.06 | 27.03 | 34.19 | 31.84 | 28.58 | 33.83 | 31.51 | 28.40 |
| Stoformer†† | 32.57 | 30.06 | 27.07 | 34.24 | 31.92 | 28.72 | 33.85 | 31.53 | 28.46 |

- Stoformer◇◇ vs. Stoformer◇†: Without the stochastic window strategy for training, directly applying the layer expectation propagation will degrade performance dramatically, which reveals that the performance gain cannot simply attribute to feature ensemble at test time and the stochastic window for training matters.

- Stoformer◇◇ vs. Stoformer†◇: Even without the layer expectation propagation algorithm for testing, the stochastic window partition for training alone is also conductive to boost performance. This is reasonable since a large amount of local information lost by Stoformer◇◇ is re-exploited by Stoformer†◇.

- Stoformer†† vs. Others: Equipped with the stochastic window for training and layer expectation propagation for testing (Stoformer††), the model achieves the highest performance.

## 4.6 Analytic Experiments

**Complexity Analysis.** We provide a detailed analysis about the time and space complexity of the stochastic window strategy based on a single attention layer. Specifically, we compare our stochastic window strategy with fixed and sliding window strategy for training and testing, respectively. For training, our strategy is as efficient as the fixed window for both speed and memory cost, as shown in Table 5. But the fixed window loses translation invariance and locality. The sliding window can fulfill the translation invariance and locality while suffer from huge memory burden, which often incurs OOM. In contrast, the stochastic window enjoys efficient training while maintains translation invariance and locality. For testing, as shown in Table 6, the fixed window can inference efficiently but suffer from the broken translation invariance and locality loss. Compared with the sliding window, our stochastic window strategy requires more computations but less memory cost.

Table 5: Time and space complexity for training. $B$: batch size, $(H, W, C)$: feature size, $s$: local window size, $h$: number of heads.

| Strategy | Time Complexity | Space Complexity | Trans. Inva. & Locality |
|---|---|---|---|
| Fixed window | $\Theta(BHWCs^2 + BHWC^2)$ | $\Theta(BHWC + BHWhs^2)$ | ✗ |
| Sliding window | $\Theta(BHWCs^2 + BHWC^2)$ | $\Theta(BHWCs^2 + BHWhs^2)$ | ✓ |
| Stochastic window | $\Theta(BHWCs^2 + BHWC^2)$ | $\Theta(BHWC + BHWhs^2)$ | ✓ |

Table 6: Time and space complexity for testing. $B$: batch size, $(H, W, C)$: feature size, $s$: local window size, $h$: number of heads.

| Strategy | Time Complexity | Space Complexity | Trans. Inva. & Locality |
|---|---|---|---|
| Fixed window | $\Theta(BHWCs^2 + BHWC^2)$ | $\Theta(BHWC + BHWhs^2)$ | ✗ |
| Sliding window | $\Theta(BHWCs^2 + BHWC^2)$ | $\Theta(BHWCs^2 + BHWhs^2)$ | ✓ |
| Stochastic window | $\Theta(BHWCs^4 + BHWC^2)$ | $\Theta(BHWC + BHWhs^2)$ | ✓ |

**Eliminating Blocking Artifacts.** Since the fixed window strategy implies infinite preference towards specific window partition, the learned feature map will consequently contain annoying blocking artifacts. In contrast, the proposed stochastic window strategy treats all window partitions equally, facilitating to keep translation invariance and model locality. Therefore, the blocking artifacts should be significantly eliminated by the stochastic window strategy. Figure 6 provides experimental evidence to support our inference. Specifically, the learned feature map by the fixed window strategy (Figure 6(b)) contains obvious blocking artifacts while the stochastic window strategy (Figure 6(c)) can remove these side effects as expected. It is surprising that with the stochastic window strategy, the feature is divided into non-overlapping local windows for further processing but in turn nearly no blocking artifacts are preserved. Please refer to the supplemental material for more visualization.

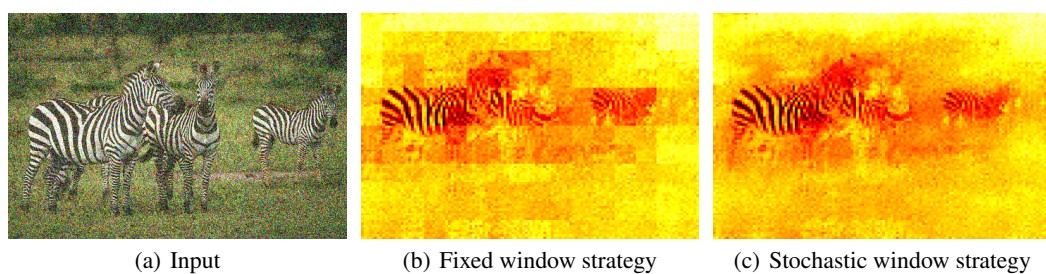

(a) Input  (b) Fixed window strategy  (c) Stochastic window strategy

Figure 6: The stochastic window strategy can eliminate the blocking artifacts.

**Trade-off in Layer Expectation Propagation.** Although the layer expectation propagation accelerates inference significantly, the strict conduction still requires to take average of $s \times s$ features yielded by self-attention. Here, we investigate the influence of the number of averaged features to model performance. Figure 7 shows the trend of model performance and *FLOPs of testing* [3] on a $256 \times 256$ image with respect to the number of averaged features each layer. Figure 7 reveals that PSNR first rises rapidly and then tends to saturate with the increasing of the averaged feature number while FLOPs increases linearly with that number, which provides a promising trade-off between performance and FLOPs. For instance, we can decrease the averaged feature number from $64$ to $8$ to keep relatively high performance and efficient inference as well.

**Boosting Model Generalization.** Figure 8 presents the loss curve on SPA-Data during training and the PSNR curve on the test set is plotted in Figure 9. Compared with the fixed window strategy, our proposed stochastic window strategy attains higher PSNR on the test set with similar training error, which demonstrates that the stochastic window strategy boosts model generalization. This can be anticipated since the stochastic window strategy can exploit complete local relationships, which are discarded by the fixed window strategy.

## 5 Limitation and Discussion

In this work, we propose the stochastic window strategy and extensive experiments validate the effectiveness on several image restoration tasks, including image deraining, denoising and deblurring. Our experiments are mainly based on the widely-used U-shaped architecture. We will further validate the stochastic strategy on more architectures (e.g., isotropic and multi-stage architecture). Indeed, the performing improvements can be positively anticipated due to the desired translation invariance and

---

[3]Note that the transformer with the stochastic window strategy can be trained as efficiently as the fixed window strategy.

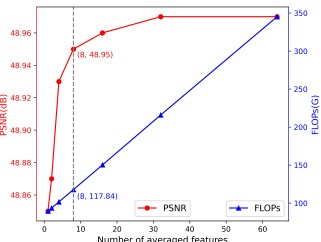

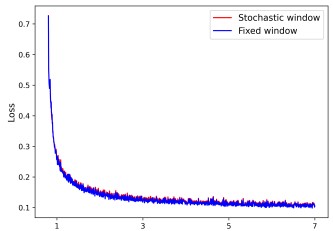

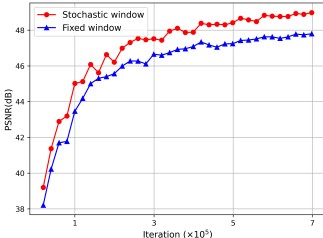

Figure 7: Trade-off between performance and FLOPs of testing.

Figure 8: Training loss on the training set of SPA-Data.

Figure 9: PSNR score on the test set of SPA-Data.

locality. Since the importance of translation invariance and locality is quite general, we also plan to extend the stochastic window strategy based transformer to more low-level vision tasks, such as image super-resolution, JPEG deblocking, and video restoration. Local attention based transformers, e.g., Swin Transformer [33], have achieved remarkable success on several CV tasks, including image classification, dense prediction, and semantic segmentation. However, they also compute attention based on specific local windows and cannot treat all the local windows fairly. The consequences are that unexpectable information loss occurs during feature processing and the translation invariance is also absent. Our proposed stochastic window strategy can provide an effective mechanism towards this problem. Therefore, it is also promising to extend the strategy to other tasks where local attention based transformers have exhibited competitive performance.

## 6 Conclusion

In this paper, we analyze the deficiencies of existing transformers for image restoration. We figure out that the fixed window strategy will inevitably lead to the broken translation invariance and loss of locality. To tackle with these issues, the novel stochastic window strategy is proposed. Specifically, we utilize the window partition with stochastic shift to replace the fixed window partition for training, which can ensure the translation invariance and intact locality. For testing, we propose the efficient layer expectation propagation to approximately take expectation of the introduced stochastic shift. Based on this strategy, we conduct extensive comparison experiments on various image restoration tasks to validate the effectiveness of the proposed stochastic window strategy.

## Broader Impacts

Nowadays, image acquisition system inevitably suffers from various degradation, ranging from inherent noise of capturing instruments, the shaking during shooting, to unpredictable weather condition. Hence, image restoration itself has important value of research and application. Our proposed stochastic window strategy can make transformer more sophisticated in restoring degraded images. However, from a societal point of view, negative consequences may also come along. For example, the deviation from the actual image textures caused by image restoration technology may affect fair judgment in medical and criminal situations. In these cases, it is necessary to combine expert knowledge to make rational decisions.

## Acknowledgement

This work was supported by the National Key R&D Program of China under Grant 2020AAA0105702, the National Natural Science Foundation of China (NSFC) under Grants 62225207 and U19B2038, the University Synergy Innovation Program of Anhui Province under Grant GXXT-2019-025, and the USTC Research Funds of the Double First-Class Initiative under Grant YD2100002003.

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
