# Supplemental Material for Stochastic Window Transformer for Image Restoration

**Jie Xiao, Xueyang Fu***, Feng Wu, Zheng-Jun Zha**
University of Science and Technology of China, Hefei, China
ustchbxj@mail.ustc.edu.cn, {xyfu,fengwu,zhazj}@ustc.edu.cn

## 1   More Details

**Small patch with global attention leads to the broken translation invariance and loss of locality.**
Due to the quadratic complexity with respect to the input resolution which is usually high for image restoration tasks, global attention applies only to the small patch in practice (e.g., $48 \times 48$ for IPT [2]). Under this setting, the broken translation invariance derives from two aspects. First, the absolute position encoding makes each token unique, which destroys the translation invariance [5, 4]. Second, the process of dividing small patches also incurs the broken translation invariance, which is similar to the aforementioned case of the fixed window partition. In a similar way, the dividing process also leads to the tremendous loss of locality.

**Complexity of the sliding window strategy.** For the sliding window strategy, every query corresponds to the distinct set of values and keys (Fig. 1(b)). This distinguishes from the fixed window strategy, in which all queries of the local window have the same set of values and keys (Fig. 1(a)). A direct consequence of the distinct context for every query is the huge memory overhead. Specifically, suppose the resolution of feature is $(H, W)$ and the size of local window is $s \times s$, the memory footprint of K and V tensor for the sliding window strategy will be $s^2$ multiple of that for the fixed window strategy (compare Algorithm 1 with Algorithm 2). Since the resolution is often high for image restoration tasks, the huge memory usage often leads to the Out of Memory (OOM) problem in practice. Furthermore, as pointed out in [13, 9], although computational complexity is $\Theta(HW)$, the sliding window strategy is still slower in wall-clock time due to the lack of optimized kernels on various accelerators.

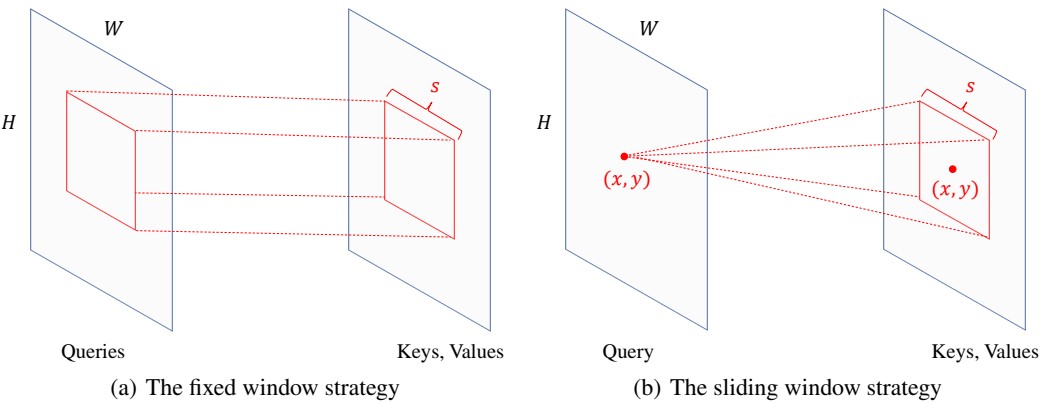

(a) The fixed window strategy      (b) The sliding window strategy

Figure 1: The illustration of context of the shifted window strategy and sliding window strategy.

---

*Corresponding author.

36th Conference on Neural Information Processing Systems (NeurIPS 2022).

**Algorithm 1** Pytorch Implementation of the fixed window based attention

```python
import torch.nn.functional as F
def FixedWindowAttention(x, win_size):
      C = x.shape[-1]
      x = F.unfold(x, kernel_size=win_size,stride=win_size)
      q, k, v = to_qkv(x)
      q = q * (C ** -0.5)
      attn = (q @ k.transpose(-2, -1))
      attn = softmax(attn + relative_position_bias)
      out = attn @ v
      return out
```

**Algorithm 2** Pytorch Implementation of the sliding window based attention

```python
import torch.nn.functional as F
def SlidingWindowAttention(x, win_size):
      C = x.shape[-1]
      q, k, v = to_qkv(x)
      k, v = pad(k), pad(v) # pad for keeping shape
      k = F.unfold(k, kernel_size=win_size, stride=1)
      #extra memory cost(win_size^2 X)
      v = F.unfold(v, kernel_size=win_size, stride=1)
      q = q * (C ** -0.5)
      attn = (q @ k.transpose(-2, -1))
      attn = softmax(attn + relative_position_bias)
      out = attn @ v
      return out
```

**Taking expectation boosts performance.** For simplicity, we denote $\{(\xi_h^l, \xi_w^l)\}_{l=0}^{N-1}$ collectively by $\xi$. We utilize the square of $L^2$ norm as the criterion to evaluate the fitted network. Therefore, given the degraded image $x$, the expected loss of the fitted $F(x, \xi)$ is given by

$$\mathbb{E}_{\xi}[||F(x,\xi) - I(x)||_2^2] \tag{1}$$

$$= \mathbb{E}_{\xi}[||F(x,\xi) - \mathbb{E}_{\xi}[F(x,\xi)] + \mathbb{E}_{\xi}[F(x,\xi)] - I(x)||_2^2] \tag{2}$$

$$= \mathbb{E}_{\xi}[||F(x,\xi) - \mathbb{E}_{\xi}[F(x,\xi)]||_2^2] + ||\mathbb{E}_{\xi}[F(x,\xi)] - I(x)||_2^2 \tag{3}$$

$$\geq ||\mathbb{E}_{\xi}[F(x,\xi)] - I(x)||_2^2. \tag{4}$$

$I(x)$ is the ground-truth image of $x$. Derivation from (2) to (3) follows that

$$\mathbb{E}_{\xi}[< F(x,\xi) - \mathbb{E}_{\xi}[F(x,\xi)], \mathbb{E}_{\xi}[F(x,\xi)] - I(x) >] = 0. \tag{5}$$

$< \cdot, \cdot >$ is the inner product. Hence, we can readily draw the conclusion that taking expectation of the introduced stochastic shift, which corresponds to $\mathbb{E}_{\xi}[F(x,\xi)]$, helps to boost performance.

## 2 Experimental Setting

**Image deraining.** We train Stoformers using two Nvidia 3090 GPUs with batch size 8 on $256 \times 256$ image pairs. The training process lasts for 10 epochs. Following previous works [15, 18], We evaluate PSNR [7] and SSIM [17] based on the luminance channel, i.e., Y channel of YCbCr space.

**Image denosing.** Following [20], we construct a large dataset comprising 400 BSD images [3], 4, 744 Waterloo Exploration Database images [10], 900 DIV2K images [1] and 2, 750 Flick2K images [8] for training. To tackle with a range of noise levels, the training images are corrupted by Gaussian noise with $\sigma$ randomly chose from $[0, 50]$. The training patches are cropped from the total training set with size $128 \times 128$. We train Stoformers using two Nvidia 3090 GPUs for total 120 epoches with batch size 16 and PSNR is evaluated on the full-size test images.

**Image deblurring.** Stoformers are trained on GoPro dataset [12][2] and directly applied to GoPro [12] and HIDE [14]. We crop $512 \times 512$ image patches with stride 256 from GoPro dataset and train

---

[2]https://seungjunnah.github.io/Datasets/gopro, CC BY 4.0 license.

Stoformers with $256 \times 256$ training pairs randomly cropped from $512 \times 512$ image patches. The total training epoch is 600 with batch size 8 on two Nvidia 3090 GPUs and we evaluate PSNR and SSIM on the full-size test images.

# 3 Visualization

## 3.1 Feature Map

Fig. 2 presents more visualization of feature maps from various depth of the stochastic window transformer and fixed window transformer. Feature maps from the fixed window transformer contain obvious blocking artifacts due to the lack of translation invariance. In contrast, the stochastic window transformer utilizes the stochastic window strategy to accomplish the translation invariance so that these artificial blocking artifacts can be removed significantly.

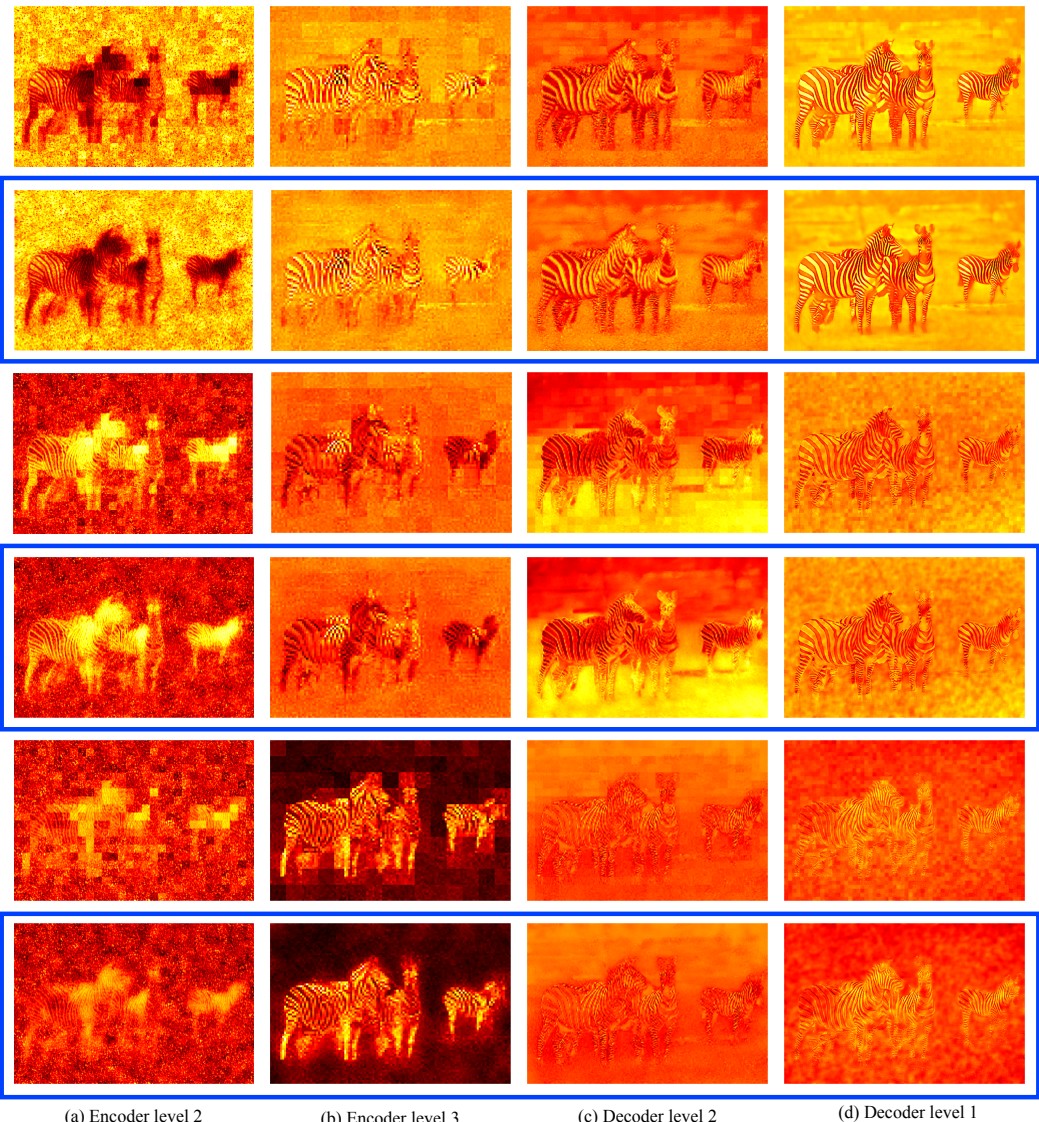

(a) Encoder level 2    (b) Encoder level 3    (c) Decoder level 2    (d) Decoder level 1

Figure 2: Feature maps from various depth of the denoising network. The feature maps in the blue box are from the stochastic window transformer while others are taken from the fixed window transformer. Please zoom in for better visualization.

## 3.2 Image Restoration Result

We also provide more visual results on image deraining (Fig. 3), image denoising (Figs. 4 and 5 for color images and Figs. 6 and 7 for grayscale images), and image deblurring (Figs. 8 and 9). In comparison with other state-of-the-art methods and Stoformer variants, Stoformer††, which is equipped with the stochastic window for training and layer expectation propagation for testing, can recover more image textures and further generate visually faithful results.

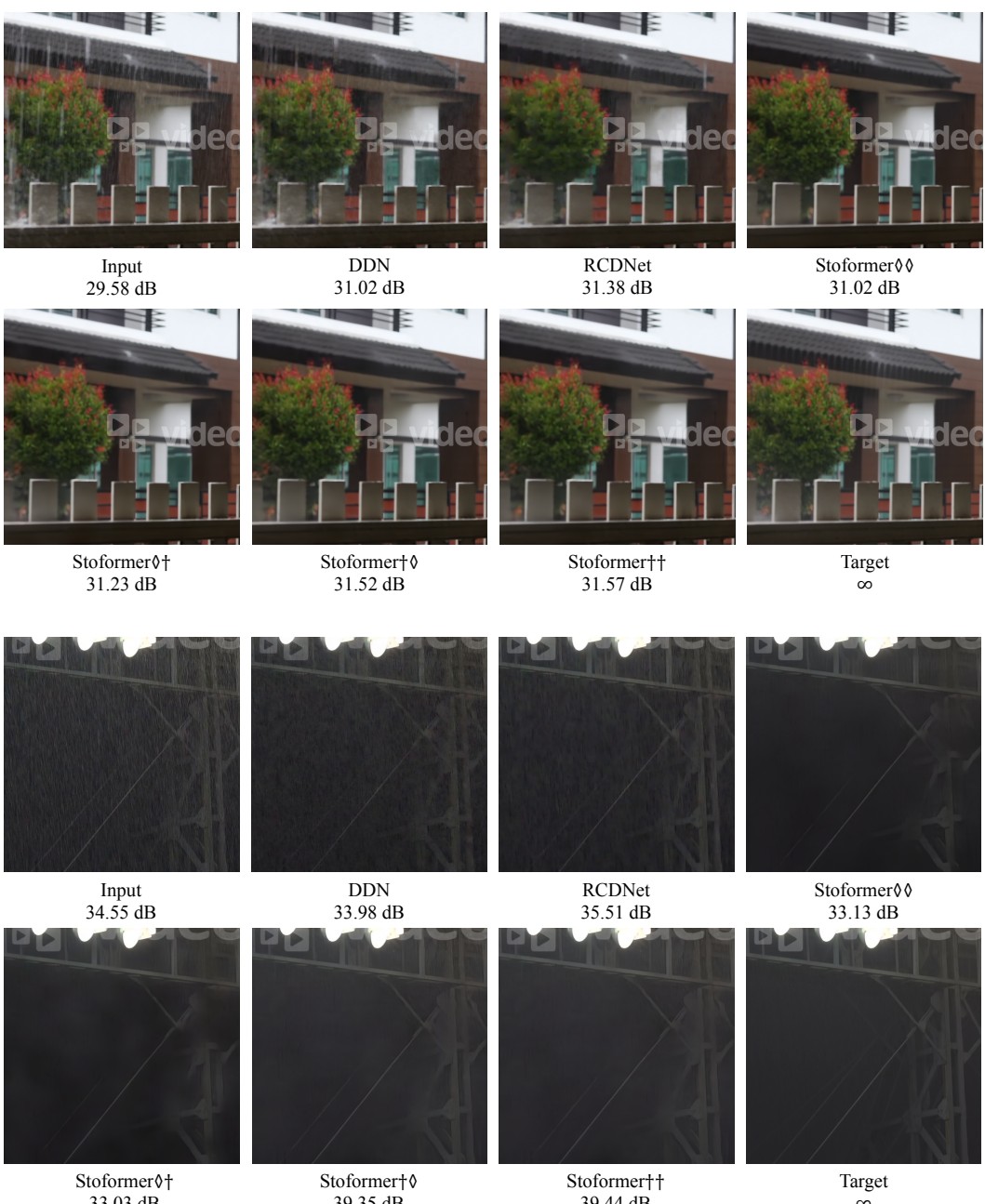

Figure 3: Visual comparison of image deraining on the SPA-Data [16].

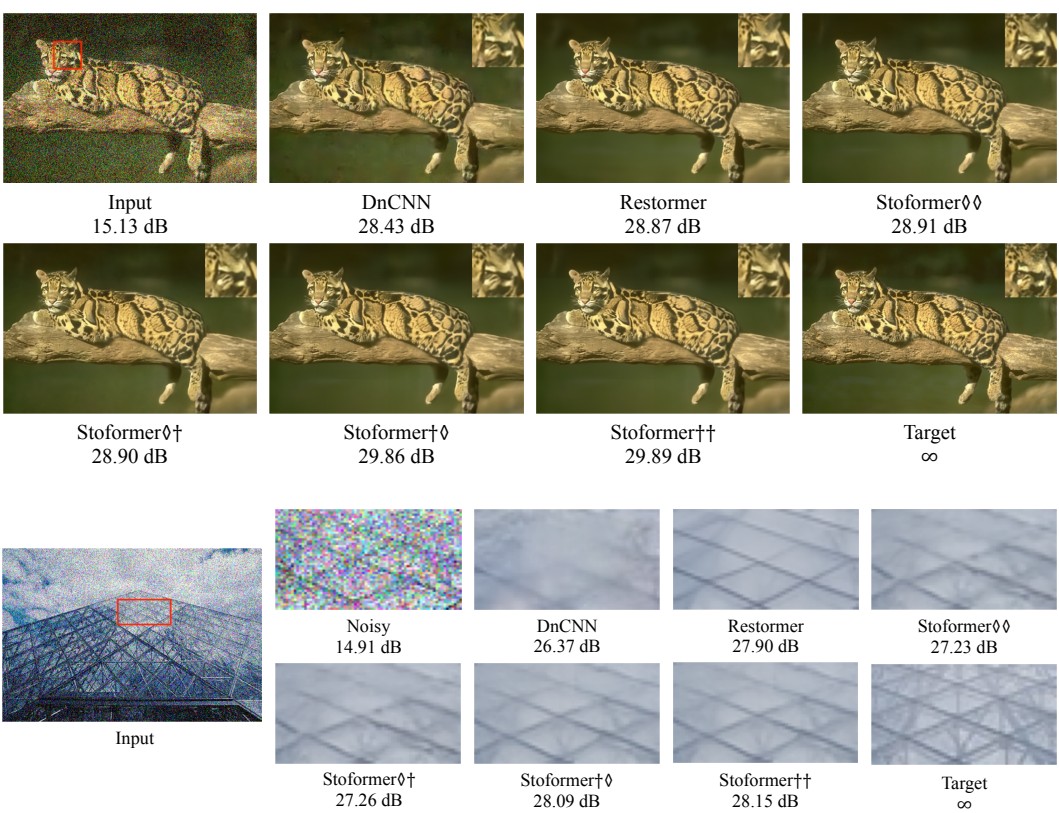

Figure 4: Visual comparison of Gaussian color denoising on the BSD68 [11].

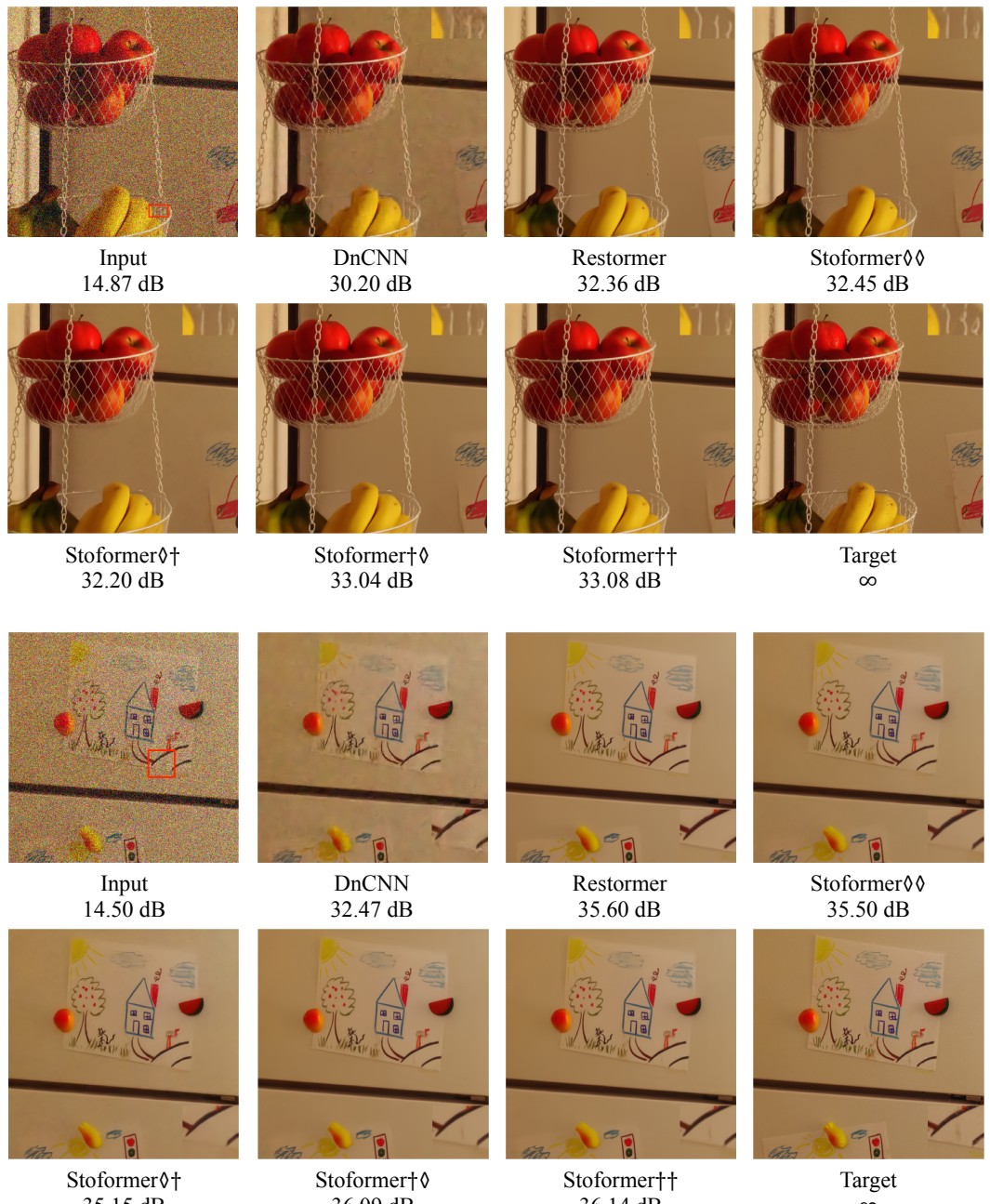

Figure 5: Visual comparison of Gaussian color image denoising on the McMaster68 [21].

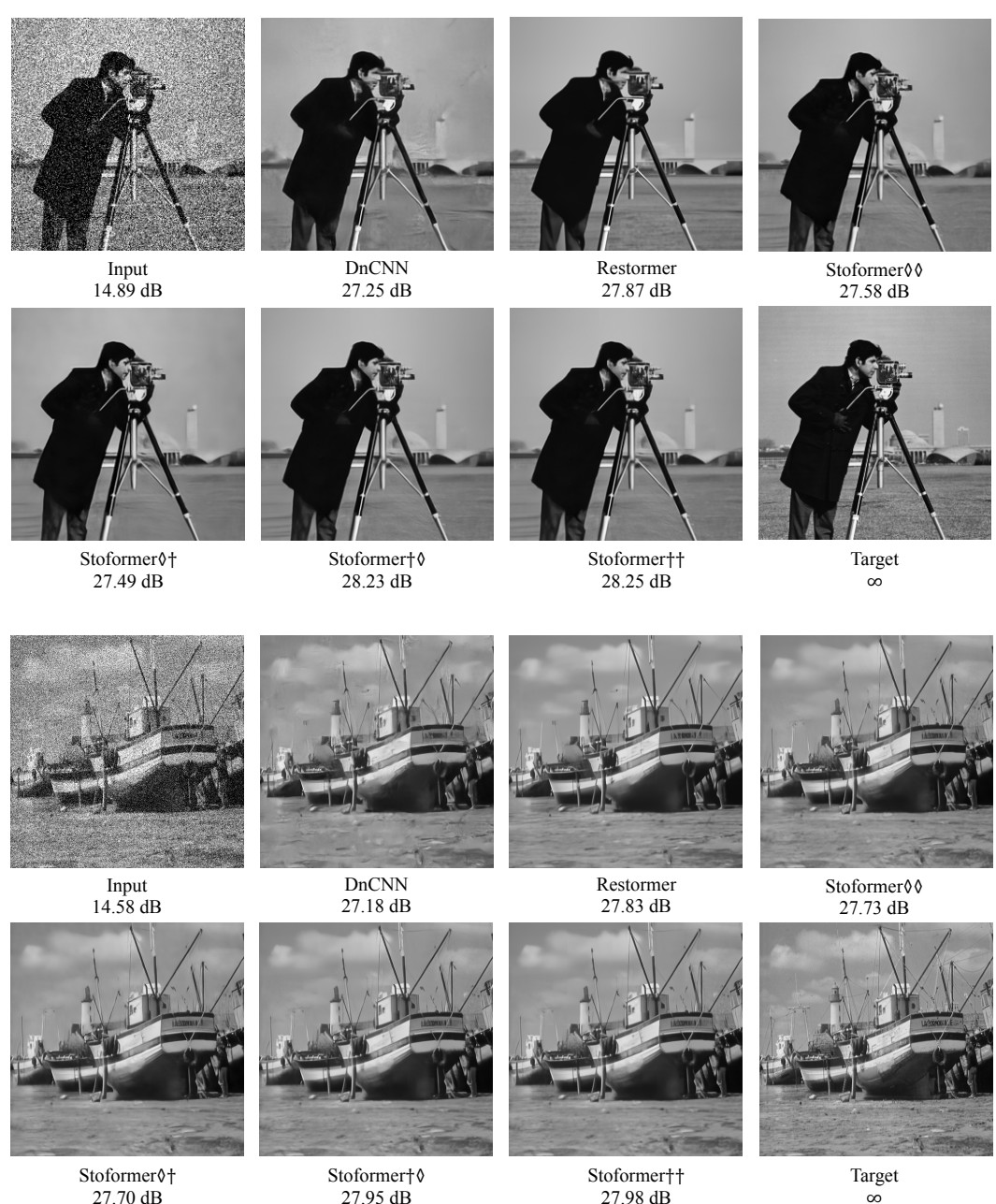

Figure 6: Visual comparison of Gaussian grayscale image denosing on Set12 [19].

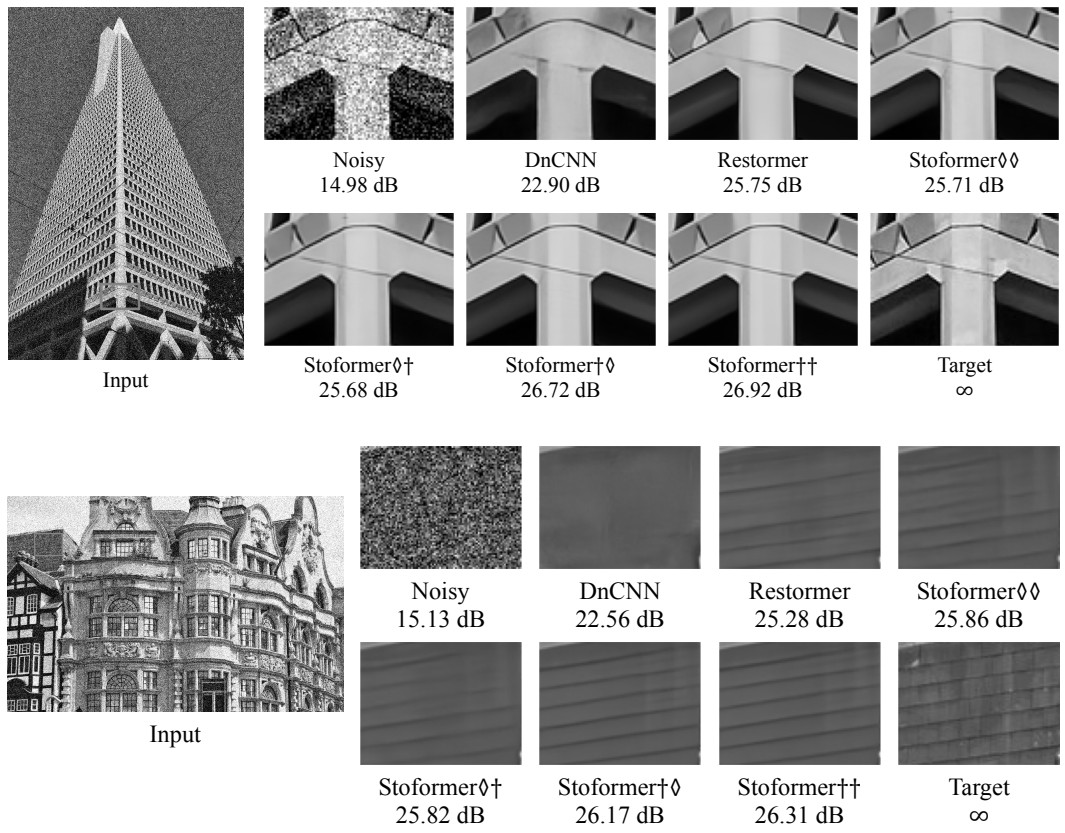

Figure 7: Visual comparison of Gaussian grayscale image denosing on Urban100 [6].

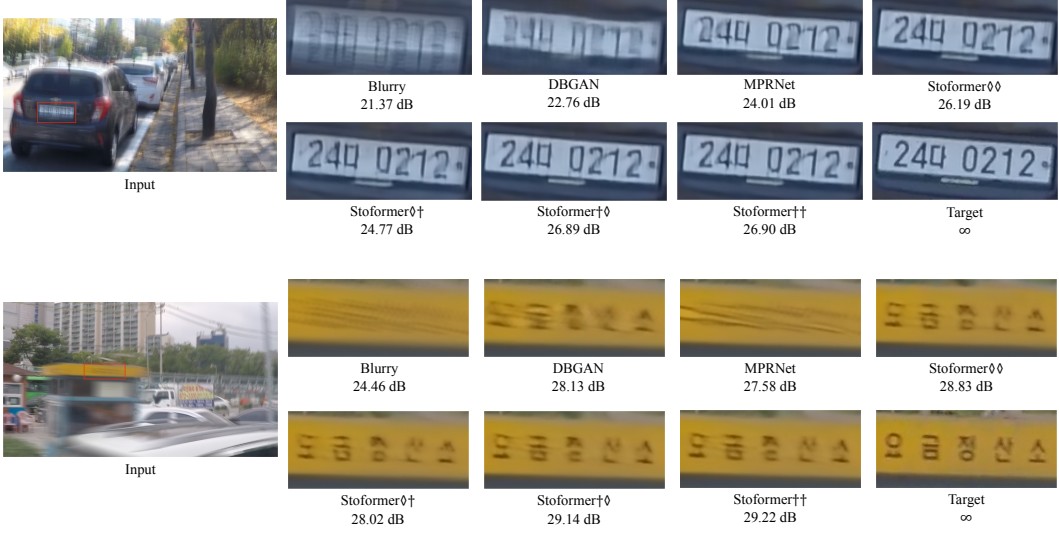

Figure 8: Visual comparison of image deblurring on the GoPro [12].

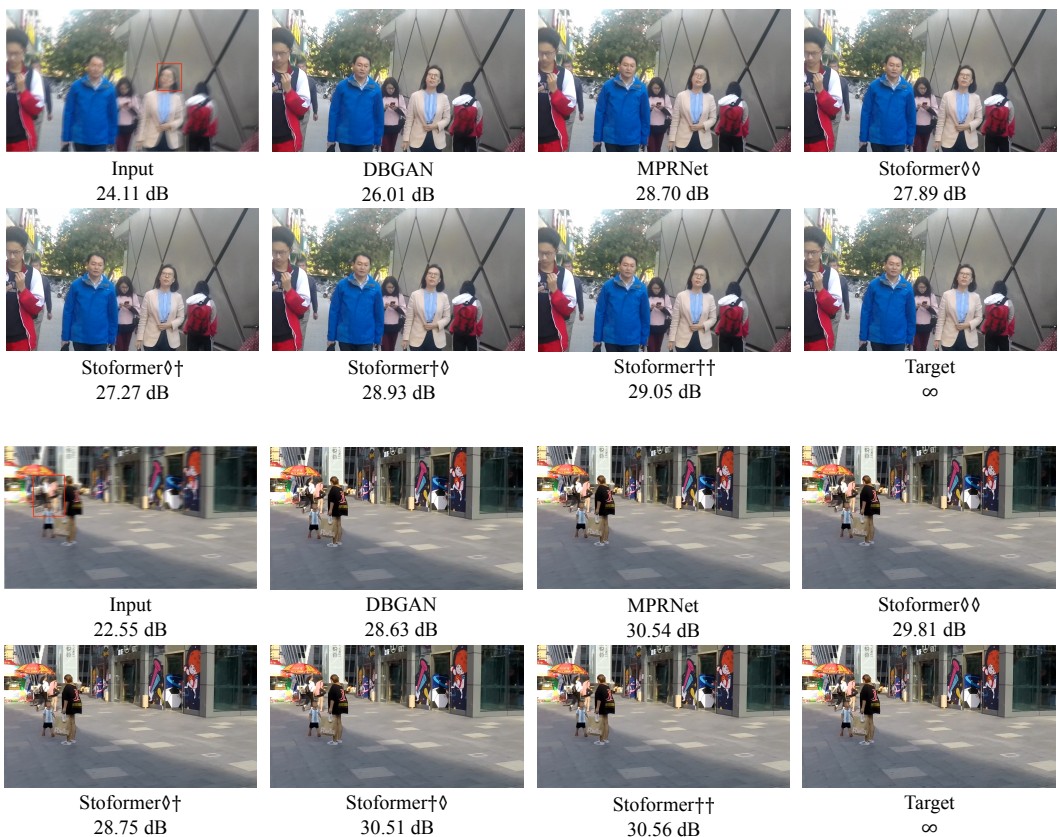

Figure 9: Visual comparison of image deblurring on the HIDE [14].