# OpenReview forum: "Stochastic Window Transformer for Image Restoration"
_NeurIPS.cc/2022/Conference — NeurIPS 2022 Accept_

### Official Review · Reviewer_jX7b · 2022-06-20

**Rating:** 8
**Confidence:** 5
**Soundness:** 4 excellent
**Presentation:** 4 excellent
**Contribution:** 4 excellent

**Summary:**

The paper points out the deficiencies, i.e., loss of locality and broken translation invariance, of existing transformers for image restoration tasks. To address these problems, the authors introduce the stochastically shifted window to replace the original fixed window with transformers and elaborate the layer expectation algorithm to take the average of corresponding stochastic factors. The paper also provides sufficient experimental evidence to support the effectiveness of the proposed method.

**Questions:**

1. See weaknesses.
2. It is suggested to release the code to encourage further research about transformers on low-level vision.


**Limitations:**

In Limitation and Future Work (Sec. 4 of S.M.), the authors point out that they will validate the proposed stochastic window strategy on more architectures and image restoration tasks. In this paper, extensive experiments across multiple degradations (image deraining, denoising, deblurring) to demonstrate that the stochastic window strategy can bring consistent improvements. Moreover, the paper also provides clear and reasonable interpretations of the improvements. Hence, it can be anticipated that the stochastic window strategy can bring improvements to other image restoration tasks and architectures. As for the potential negative societal impacts, the authors also provide several solutions to mitigate them.

**Strengths And Weaknesses:**

Strength:
1. The paper provides significant insights about applying transformers to image restoration tasks. For low-level vision, researchers often directly apply high-level design keys of transformers into low-level vision without careful considerations. This paper exhibits consequent loss of locality and broken translation invariance of transformers. Given the potentially general impact of transformers on computer vision, analyzing the impropriety of transformers on specific tasks is of great meaning.
2. The proposed method is quite novel and effective. To address the problem of the broken translation invariance and locality loss, the authors adopt the idea of sampling to uniformly sample one shift configuration (ξ_h, ξ_w) and design the efficient yet effective layer expectation propagation algorithm to take the expectation of the introduced stochastic factors. The authors also conduct extensive experiments to support the proposed method.
3. The paper is well organized and easy to be understood.

Weakness:
1. Line 144-145 says that “stochastic shifts should be averaged according to their posterior distribution” and Eq. 4 implies that these random variables are averaged according to a uniform distribution. So, there should be some explanations about why the implied posterior distribution is uniform.
2. The stochastic window transformer contains the layer expectation propagation algorithm to approximate the exact average (expectation) of stochastic shifts. It is suggested to add a concrete explanation about the reason of taking expectations.

---

> ### Author Response · Authors · 2022-08-02
> **Response to Reviewer jX7b**
>
> **Q1:** Explanations about why the implied posterior distribution is uniform.
>
> **A1:** In this work, we discard infinite favoritism towards the certain partition and propose to exploit all the local windows fairly. In order to achieve this goal, we assume the posterior distribution is uniform and stochastic shifts are sampled from uniform distribution independently and identically.
>
> **Q2:** Adding a concrete explanation about the reason of taking expectations.
>
> **A2:** The reason we take expectation of the stochastic shifts can be two-fold. First, taking expectation is proven to consistently improve the model performance in the SM (L48-53). Hence, taking expectation is  favorable. Second, we need to maintain the translation invariance and intact locality during testing. Taking expectation means all the local windows are treated fairly so that the translation invariance and intact locality can be ensured. However, as we analyze in the paper, naive implementation will incur unacceptable computational burden. To this issue, we propose to take layer-wise expectation to replace original expectation, in which computational complexity reduces from $\Theta(s^{2N})$ to $\Theta(Ns^2)$. Besides, the layer expectation propagation still maintains the desired translation invariance and locality.
>
> **Q3:** It is suggested to release the code to encourage further research about transformers on low-level vision.
>
> **A3:** We will release the code to encourage reproducibility.

---

> > ### Comment · Reviewer_jX7b · 2022-08-05
> > **After rebuttal**
> >
> > I appreciate your responses. All of my concerns have been addressed by the response.
> >
> > In my opinion, this work is interesting since it analyzes the drawbacks of transformer, including broken translation invariance and local information loss, and proposes the novel stochastic window strategy to effectively solve this problem.
> >
> > Moreover, I like the toy examples in Figure 4, which visualizes the restored translation invariance of the stochastic window strategy.
> >
> > Other reviewers mention the issue of complexity and the authors provide sufficient comparisons with other strategies in rebuttal. I believe this work is novel enough and pushes the development of transformers. After seeing the rebuttal and other review comments, I still insist on my score or consider arising the score.

---

> > > ### Author Response · Authors · 2022-08-08
> > > **Thanks**
> > >
> > > Thank you sincerely for your positive comments and suggestions.

---

### Official Review · Reviewer_86x1 · 2022-07-11

**Rating:** 6
**Confidence:** 3
**Soundness:** 3 good
**Presentation:** 3 good
**Contribution:** 3 good

**Summary:**

This paper is interesting. Spatial attention is essential in windowed transformer. By stochastic varying the window sizes and shapes, image restoration benefits from signficantly in various testsets compared with SOTAs.

**Questions:**

1. Can this technique be extended to other CV tasks ?

**Limitations:**

1. Failure cases can be presented.

**Strengths And Weaknesses:**

Strengths:
1. First work of Stochastic Window transformer for image restoration
2. New SOTAs for image restoration
Weaknesses:
1. Complexity of new stochastic window is not practically analyzed.
2. Improvement on visual quality is not very obvious.

---

> ### Author Response · Authors · 2022-08-02
> **Response to Reviewer 86x1**
>
> **Q1:** Complexity of new stochastic window is not practically analyzed.
>
> **A1:** We provide a detailed time/space complexity analysis for a single attention layer. Specifically, we compare our strategy with the fixed and sliding window strategy for training and testing, respectively. We adopt notations: $B$: batch size, $(H, W, C)$: feature size, $h$: number of heads, $s$: local window size.
>
> Table 1: Time and space complexity for training.
> |Strategy|Time Complexity|Space Complexity|Trans. Inva.|Locality|
> |:---- |:----:|:---:|:---:|:---:|
> |Fix. Win.|$\Theta(BHWCs^2+BHWC^2)$|$\Theta(BHWC+BHWhs^2)$|False|False|
> |Slid. Win.|$\Theta(BHWCs^2+BHWC^2)$|$\Theta(BHWCs^2+BHWhs^2)$|True|True|
> |Sto. Win.|$\Theta(BHWCs^2+BHWC^2)$|$\Theta(BHWC+BHWhs^2)$|True|True|
>
> For training, our strategy is as efficient as the fixed window for both speed and memory cost, as shown in Tab. 1. But the fixed window loses the translation invariance and locality. The sliding window can fulfill the translation invariance and locality while suffer from huge memory burden, which often incurs OOM in practice. In contrast, the stochastic window enjoys efficient training while maintains translation invariance and locality.
>
> Table 2: Time and space complexity for testing.
> |Strategy|Time Complexity|Space Complexity|Trans. Inva.|Locality|
> |:----|:----:|:---:|:---:|:---:|
> |Fix. Win.|$\Theta(BHWCs^2+BHWC^2)$|$\Theta(BHWC+BHWhs^2)$|False|False|
> |Slid. Win.|$\Theta(BHWCs^2+BHWC^2)$|$\Theta(BHWCs^2+BHWhs^2)$|True|True|
> |Sto. Win.|$\Theta(BHWCs^4+BHWC^2)$|$\Theta(BHWC+BHWhs^2)$|True|True|
>
> For testing, it requires to exploit all the local windows for a single forward pass so that extra cost is required. As shown in Tab. 2, the fixed window can inference efficiently but suffer from the broken translation invariance and locality loss. Compared with the sliding window, our stochastic window strategy requires more computations but less memory cost. We have added the complexity analysis to the SM (L54-63).
>
> **Q2:** Improvement on visual quality is not very obvious.
>
> **A2:** Indeed, visual improvement can be more obvious within regions with more textures. We have added markers in Blurry of to assist to locate some of these regions. Please refer to the revision (Page 7) and visual improvement can be easier to be captured.
>
> **Q3:** Can this technique be extended to other CV tasks?
>
> **A3:** Local attention based transformers, e.g., Swin Transformer [1], have achieved remarkable success on several CV tasks, including image classification, dense prediction, and semantic segmentation. However, they also compute attention based on specific local windows and cannot treat all the local windows fairly. The consequences are that unexpectable information loss occurs during feature processing and the translation invariance is also absent. Our proposed stochastic window strategy can provide an effective mechanism towards this problem. Therefore, it is very promising to extend the strategy to other tasks where local attention based transformers have exhibited competitive performance. Indeed, except for other restoration tasks mentioned in Limitation and Future Work, we are also considering extending this strategy to other CV tasks, such as image classification, dense prediction and semantic segmentation.
>
> **Q4:** Failure cases can be presented.
>
> **A4:** In this paper, we propose the stochastic window strategy to recover the broken translation invariance and locality loss. Extensive experiments and analyses have demonstrated the strategy can accomplish this goal and improve performance across multiple tasks. We cannot observe and forecast failure cases from the sense of violating the translation invariance and intact locality. But for restoring clean images, there are some cases where the degenerations are so heavy that our strategy cannot remove thoroughly. Please refer to the SM (L95-96 and Fig. 10) for failure cases.
>
> [1] Swin transformer: Hierarchical vision transformer using shifted windows. ICCV, 2021.

---

> > ### Author Response · Authors · 2022-08-08
> > **More responses**
> >
> > We appreciate for your comments. We hope that this response can address your concerns. We are delighted to provide more details if needed.

---

### Official Review · Reviewer_KzwD · 2022-07-11

**Rating:** 3
**Confidence:** 4
**Soundness:** 2 fair
**Presentation:** 2 fair
**Contribution:** 2 fair

**Summary:**

This paper raises an issue regarding the loss of translation equivariance in image restoration problems due to the transformer-based architectures. Often, attention or MLP layers operate on a predetermined window locations and the authors argue this could cause undesired artifacts or performance drops.
The authors proposes Stochastic Window Transformer by 1) allocating random window positions at training time and 2) averaging the results from multiple window positions.
The experiments were done in image deraining, denoising, and deblurring.

**Questions:**

I would suggest the authors check the definition of translation-equivariance and translation-invariance.
Also, the proposed method does not guarantee translation-equivariance.

Please refer to the weaknesses.

**Limitations:**

Limitations were addressed by the authors.

**Strengths And Weaknesses:**

- translation equivariance

translation invariance -> translation equivariance
As translation equivariance is the key property this paper is addressing, the authors should use a proper terminology.
An example task where translation invariance is needed is image classification, recognizing a number correctly whether it is shifted or not.
Given a translation function T,
Translation invariance: f(T(x)) = f(x)
Translation equivariance: f(T(x)) = T(f(x))

L29 is the desiderata -> is one of the desiderata
I would omit ‘any’ to be safe. There could be cases the position is important, ex) recovering from image distortion, vignetting, etc.

L39-40 As attention layers are involved, I don’t expect the method to be translation equivariant in any ways.

- locality

L36, Figure 1b. If non-overlapping patches cause problems in locality, we could always use overlapping patches and crop out the boundaries at the cost of additional computation. It could be less efficient but solves the problem. Such strategy is commonly used to process high-resolution images even for translation-equivariant methods (CNNs).

- layer expectation propagation algorithm

How does equation (5) work? Does it require inference over xi to compute expectation per attention block?
This introduces s^2 times more inference computations. Is the result worth the computation?
While Figure 7 claims reducing the number of inference brings efficiency, however, x8 inference is still a very large burden.
Also, in the supplementary material, the PSNR differences between the inference results with and without the layer expectation propagation are minor.


- Figure 4

What is shown in Figure 4?
What does the z-axis mean?
L211 Is the PSNR normalization factor the same across 4b, 4c, and 4d?
Translation equivariance is one of the desired property but achieving high PSNR is also important.
Which approach achieved the best average PSNR?

Why want translation equivariance in transformers? People use transformers by sacrificing equivariance to achieve better results. Does bringing approximate translation equivariance lead to higher accuracy?

- Figure 5

While the PSNR gains in the Stoformer++ is big, the visual result doesn’t look that much different from other results.
I don’t find this example to be appealing.

- References

[30] It is ICCV Workshop paper, not ICCV

- Typos and Grammar

L17 constantly -> consistently
L32 What does ‘concluded’ mean? Is it ‘considered’?
L167 By contrast -> In contrast

---

> ### Author Response · Authors · 2022-08-02
> **Response to Reviewer KzwD**
>
> **Q1:** The authors should use translation equivariance rather than translation invariance.
>
> **A1:** First, we would like to re-emphasize the translation invariance is actually what we delve into. As we have clearly stated in the submisison, the ideal image restoration method should be invariant to the translation of degradation. In other words, the algorithm should be immune to the position of degeneration (e.g., rain streaks, noise). Degeneration concerned in image restoration is analogy with the number whose position can be shifted in image classification, and the restored image corresponds to the recognized label. *Therefore, there is no doubt that in this paper, the use of translation invariance to describe the problem of our study is correct.* **L29** is not unique.  We have modified it as one of the desiderata to avoid confusion. Translation invariance is also required for image distortion and vignetting, since ideally the same clean image is returned. **L39-40.** We are discussing about translation invariance not equivariance. Due to weight sharing, translation invariance among paritioned windows happens [1,2].
>
> **Q2:** Overlapping patch can solve the problem locality.
>
> **A2:** Fig. 1b intuitively illustrates the loss of local relationships brought by the fixed window partition. The non-overlapping partition of feature is conducted for *calculating local attention*. Note that partition here serves for *calculating local attention rather than processing high-resolution images.* Overlapping partition cannot solve the locality problem unless the overlapping configuration is as compact as the sliding window, which is demanding for computational resources [5,6]. Most local attention based transformers [1,2,3,4] adopt non-overlapping rather than overlapping partition for the issue of efficiency. Besides, CNNs have efficient implementation based on the sliding window so that partition (overlapping or non-overlapping) for feature processing is unnecessary. Consequently, locality loss that comes with the fixed window partition is a non-trivial issue. Our strategy can solve the locality problem elegantly.
>
> **Q3:** Layer expectation propagation (LEP) algorithm.
>
> **A3:** Eq. (5) requires inference over $\xi$ to calculate the expectation per attention block, which incurs extra computations. However, the extra computations can bring: 1) lower memory cost. LEP is proposed to approximately take expectation of the introduced stochastic factors. Hence, LEP requires to process all the local windows for a single forward pass. Compared with the sliding window, which also achieves the function, LEP requires more computations but less memory (please refer to L54-63 of the SM). 2) consistent performance gains. Despite being relatively smaller, the performance gains can be consistently obtained. 3) translation invariance and locality. For training, we propose to implement the translation invariance and locality using stochastically shifted window. For testing, we resort to LEP to maintain the above two functions.
>
> **Q4:** What is shown in Fig. 4?
>
> **A4:** In Fig. 4, we use the toy experiment to validate that our stochastic window strategy can achieve the translation invariance. As shown in Fig. 4 (a), when we shift the added Gaussian noise patch with offset $(\xi_h, \xi_w)$, the model with translation invariance should produce the same recovered result so the PSNR distribution w.r.t. offset should be as flat as possible. PSNR is normalized to [0, 1] because we aim to visualize variations of PSNR w.r.t. offset. Extensive experiments are conducted in Sec. 4.3-4.5 to demonstrate the superior performance of our strategy. In Fig. 4, we visualize the broken translation invariance of fixed window based transformer and our strategy can yield relatively uniform outputs like CNNs-based VDN.
>
> **Q5:** Visual improvement is not very obvious in Fig. 5.
>
> **A5:** Indeed, visual improvement can be observed within regions with more textures. We have added markers in Blurry of Fig. 5 to assist to locate some of these regions. Please refer to Fig. 5 of the revised vision and visual improvement can be easier to be captured.
>
> **Q6:** References & Typos & Grammar.
>
> **A6:** We have carefully polished the whole manuscript in the revision.
>
> [1] Swin transformer: Hierarchical vision transformer using shifted windows. ICCV, 2021.
>
> [2] Swin transformer v2: Scaling up capacity and resolution. CVPR, 2022.
>
> [3] Hrformer: High-resolution vision transformer for dense predict. NeurIPS, 2021.
>
> [4] Twins: Revisiting the design of spatial attention in vision transformers. NeurIPS, 2021.
>
> [5] Stand-alone self-attention in vision models. NeurIPS, 2019.
>
> [6] Local relation networks for image recognition. ICCV, 2019.

---

> > ### Author Response · Authors · 2022-08-08
> > **More responses**
> >
> > Thank you for taking the time to review our work. We would be glad to provide more details if you have any question.

---

> > > ### Comment · Reviewer_KzwD · 2022-08-09
> > > **Post-rebuttal response**
> > >
> > > I thank the authors for their rebuttal and detailed explanations.
> > > Here are post-rebuttal responses.
> > >
> > > 1. Translation invariance vs translation equivariance.
> > >
> > > I am pretty much confident that translation invariance is what we don't usually want in image restoration literature.
> > > Translation invariance refers to the property where the output of a model does not change by the translation of input.
> > > Translation equivariance refers to the property where the output of a model changes by the same amount of translation as the input.
> > > If an image restoration algorithm is translation invariant, it means that it will always return the same output regardless of the input, making the algorithm useless.
> > > I highly recommend the authors check the definition of basic concepts. I can easily find many materials explaining the differences between invariance and equivariance from a quick search on the web.
> > > ex) https://www.doc.ic.ac.uk/~bkainz/teaching/DL/notes/equivariance.pdf
> > >
> > > Note: I don't mean translation equivariance is what we always want. I'm trying to explain translation invariance is not usually a desired property.
> > >
> > > I am concerned that the authors are abusing the term invariance by referring to the degradation instead of the input.
> > > (I initially thought it was a grammar/clarity issue, but from the author response, it seems not.)
> > > However, translation in- / equi-variance are terms describing the behavior of model to its input, not to another function applied to input (here, degradation).
> > > If my understanding is correct, the authors should claim they want to build a degradation position-agnostic image restoration algorithm instead of misleading terms.
> > > One additional concern is that the authors are mixing up the two concepts (the claimed invariance to degradation position and typical translation equivariance) throughout the paper and the above response.
> > > The proposed approach is building on the translation equivariance but not on the idea of handling degradation position variation.
> > >
> > > Besides the terminology, the authors' claim is not very precise. No practical image restoration methods produce the same output from an image with different degradation positions. (except trivial solutions such as global averaging)
> > > SoTA methods do not achieve the authors' desired property is not because they adopt transformer architectures but simply because it is an extremely difficult problem (if not impossible). (L2-3, 7-9)
> > >
> > > Furthermore, to be precise, we expect that an ideal image restoration method to recover the image regardless of the degradation types, strength, direction, etc., not just limited to the position of degradation artifacts.
> > > If the authors would claim the degradation position-agnostic recovery is important, they should justify it first.
> > >
> > > 2. locality
> > > It is hard to agree that the proposed method handles the locality issue better than overlapping patches.
> > > In principle, the proposed Layer Expectation Propagation is computing the features multiple times by shifting the window.
> > > I don't see essential differences.
> > >
> > > 3. Layer Expectation Propagation algorithm
> > >
> > > 3-1. I don't understand how LEP reduces memory consumption. By referring to L54-56, are the authors comparing their method with sliding-window style inference?
> > > That is not a fair baseline.
> > > The authors aim to improve the baseline transformer architecture and the comparison should be made with the baseline model's single inference.
> > > Comparing the proposed method with an imaginary bad method is not a scientific approach.
> > > Actually, the proposed method requires multiple inferences and requires the memory to store the intermediate features.
> > > LEP potentially increases peak memory consumption, not reduces it.
> > > Did the authors measure the actual memory usage?
> > >
> > > 3-2. Performance gains from extra computation is not surprising.
> > >
> > > 4.  Figure 4
> > >
> > > I understand what the authors' intention is. Will the authors add the missing explanations and details to the manuscript?
> > >
> > > 5. Figure 5
> > >
> > > Ok.

---

> > > > ### Author Response · Authors · 2022-08-09
> > > > **Response**
> > > >
> > > > Thank the reviewer for the response. Here are our responses.
> > > >
> > > > **Q1:** Translation invariance vs translation equivariance.
> > > >
> > > > **A1:** We can comprehend the difference between invariance and equivariance. We must re-clarify that the terminology translation invariance is accurate since it reflects the invariance to shifts of the interested degradation. The reviewer's understanding of invariance and equivariance is not general enough. The reviewer believes that translation must be conducted on the whole noisy image, which is not true. The reviewer admits translation invariance is needed for image classfication. In classification, translation can be focused on interested object and the recognized label should be invariant to the shifts of the object. Therefore, translation invariance is needed in classfication. Consider the case where we want to recognize two independent digits in a image. If translation was limited on the whole image, it means that translation invariance could only be ensured by joint shifts of the digits. Obviously, it is not reasonable because translation invariance should allow to return the same results regradless of *independent shifts* of the digits within the image. Consequentlty, translation should be allowed conducted on particular objects. Moreover, as the given material says (Page 95), ``A network that is insensitive to shifts of the object of interest in an image is called shift invariant.`` We can also provide another evidence in the classic book [9] that supports our argument:  ``a particular object should be assigned the same classification irrespective of its position within the image (translation invariance)`` in Chapter 5.5.3 Invariances. If the reviewer can understand this, the concern can be readily solved. Here, the so-called interested object is dagradation, which is the target of image restoration algorithm. **Consequently, translation invariance is the proper terminology.** The commemt ``If an image restoration algorithm is translation invariant, it means that it will always return the same output regardless of the input, making the algorithm useless`` is apparently wrong. Because we only allow invariance to the shifts of degradation. If the image background is changed, the output is changed accrodingly.
> > > >
> > > > 1-2 Besides, we fix up the broken translation invariance of transformers for image restoration tasks. We do not mean other desired properties are not necessary. **We believe the reviewer should focus on image restoration but also transformer.** Translation invariance can be approximately accomplished by the sliding window adopted in CNN, while the sliding window strategy is too expensive in local window based transformers [5,6]. Hence, existing transformers adopt the fixed window strategy, which leads to the broken translation invariance and local information loss. Equipped with our stochastic window strategy, transformer can possess the desired translation invariance and intact locality, which is more suitable for image restoration. We have conducted extensive experiments to demonstrate that the stochastic window transformer outperforms existing fixed window strategy, i.e., shifted window transformer, for image restoration tasks.
> > > >
> > > > We have to remind the reviewer that equipped with the stochastic window strategy, transformer can be translation invariant and exploit intact local information for image restoration. These properties are desirable for image restoration tasks (other tasks, e.g., image classification, may also be suitable). Our strategy aims to eliminate the limitations of existing fixed window strategy of transformers for image restoration.
> > > >
> > > > **Q2:** Locality
> > > >
> > > > **A2:** Existing works that implement local window based attention include the sliding window strategy [5,6] and fixed window strategy [1,2,3,4,7,8]. The fixed window strategy (e.g., shiftded window strategy) is the mainstream for local attention based transformer since it is more efficient compared with the sliding window strategy. There is no ``overlapping patches and crop out the boundaries`` strategy for local attention in existing literatures because it is purely meaningless. The sliding/fixed window strategy have their own merits, i.e., the sliding window can maintain translation invariance and intact locality while the fixed window strategy is more efficient. As for ``overlapping patches and crop out the boundaries``, it is neither efficient nor keeping desired property. We remind the reviewer that the parition of feature prepares for the calculation of local attention not for procossing high-resolution images.

---

> > > > > ### Author Response · Authors · 2022-08-09
> > > > > **follow-up**
> > > > >
> > > > > **Q3:** Layer Expectation Propagation
> > > > >
> > > > > **A3:** 3-1 LEP requires the nearly same memory as the fixed window partition. For the sliding window strategy, the context for each token is different so that it requires more memory compared with the fixed window strategy. Please refer to Fig. 1 in the SM for intuitive illustration and Algorithm 2 in the SM for implementation details (The implementation derives from [5][(code)](https://github.com/MartinGer/Stand-Alone-Self-Attention-in-Vision-Models)). Besides, the sliding window strategy is the scientific approach [2,6,7] and we also compare with the fixed window strategy in our paper. We comprehensively analyze the superiority of our method, including performance improvements compared with fixed window strategy and efficiency improvements compared with the sliding window strategy. The comment ``Actually, the proposed method requires multiple inferences and requires the memory to store the intermediate features`` is not correct, since we only need to maintain the averaged feature and intermediate features are directly discarded. We have measured the actual memory usage in the A3 to Reviewer S7Qf. We copy it below for your convenience.
> > > > >
> > > > > The meaning of each symbol: $B$: batch size, $(H, W, C)$: feature size, $s$: local window size, $h$: number of heads. We also report FLOPs and memory cost on a single attention layer, in which $B=1, h=1, s=8, H=W=256, C=128.$ We use GTX 1080Ti GPU as the computing device. OOM means out of memory(>11G).
> > > > >
> > > > > Table 1: The complexity for training.
> > > > > |Strategy|Time Complexity|Space Complexity|FLOPs (G)|Memory(G)|Trans. Inva.|
> > > > > |:----|:----:|:---:|:---:|:---:|:---:|
> > > > > |Fix. Win.|$\Theta(BHWCs^2+BHWC^2)$|$\Theta(BHWC+BHWhs^2)$|5.4|1.3|False|
> > > > > |Slid. Win.|$\Theta(BHWCs^2+BHWC^2)$|$\Theta(BHWCs^2+BHWhs^2)$|5.4|OOM|True|
> > > > > |Sto. Win.|$\Theta(BHWCs^2+BHWC^2)$|$\Theta(BHWC+BHWhs^2)$|5.4|1.3|True|
> > > > >
> > > > > Table 2: The complexity for testing.
> > > > > |Strategy|Time Complexity|Space Complexity|FLOPs (G)|Memory(G)|Trans. Inva.|
> > > > > |:----|:----:|:---:|:---:|:---:|:---:|
> > > > > |Fix. Win.|$\Theta(BHWCs^2+BHWC^2)$|$\Theta(BHWC+BHWhs^2)$|5.4|1.2|False|
> > > > > |Slid. Win.|$\Theta(BHWCs^2+BHWC^2)$|$\Theta(BHWCs^2+BHWhs^2)$|5.4|OOM|True|
> > > > > |Sto. Win.|$\Theta(BHWCs^4+BHWC^2)$|$\Theta(BHWC+BHWhs^2)$|72.3|1.2|True|
> > > > >
> > > > > 3-2 Our strategy contains two parts, stochastic window for training and LEP for testing. The proposed  ``Performance gains from extra computation is not surprising`` demonstrates stochastic window for training is effective and provides a strong baseline, which is also our contribution. The LEP can even consistently outperform the strong baseline with more computations.
> > > > >
> > > > > **Q4:** Figure 4
> > > > >
> > > > > **A4:** Please refer to Sec. 4.2 where we provide explanations and details.
> > > > >
> > > > >
> > > > > [1] Swin transformer: Hierarchical vision transformer using shifted windows. ICCV, 2021.
> > > > >
> > > > > [2] Swin transformer v2: Scaling up capacity and resolution. CVPR, 2022.
> > > > >
> > > > > [3] Hrformer: High-resolution vision transformer for dense predict. NeurIPS, 2021.
> > > > >
> > > > > [4] Twins: Revisiting the design of spatial attention in vision transformers. NeurIPS, 2021.
> > > > >
> > > > > [5] Stand-alone self-attention in vision models. NeurIPS, 2019.
> > > > >
> > > > > [6] Local relation networks for image recognition. ICCV, 2019.
> > > > >
> > > > > [7] Uformer: A general u-shaped transformer for image restoration. CVPR, 2022.
> > > > >
> > > > > [8] Swinir: Image restoration using swin transformer. ICCV Workshops, 2021.
> > > > >
> > > > > [9] Bishop C M, Nasrabadi N M. Pattern recognition and machine learning[M]. 2006.

---

### Official Review · Reviewer_S7Qf · 2022-07-11

**Rating:** 3
**Confidence:** 4
**Soundness:** 3 good
**Presentation:** 3 good
**Contribution:** 2 fair

**Summary:**

This paper proposes a stochastic window strategy for training transformers and elaborates layer expectation propagation algorithm during the testing. This strategy consistently improves the performance of several image restoration tasks.

**Questions:**

See the Strengths And Weaknesses.

**Limitations:**

See the Strengths And Weaknesses.

**Strengths And Weaknesses:**

Strengths:
1. The idea is interesting. With the stochastic window strategy, attention modules can avoid the broken translation invariance.
2. The experiments show the improvements of each step clearly.

Weaknesses:
1. The training step is easy to understand. But, I do not fully understand the testing step. How to implement it to avoid the original expensive operations? Does the testing flops increase compared with the training? How about the detailed comparisons (flops and memory cost) of your method with the fixed(shift) window and sliding window during the training and testing?

2. I encourage the author to adopt this strategy to existing transformer-based methods (e.g., Uformer[1], swinIR[2]) to evaluate the generalization on different backbones.

3. L245. Without the layer expectation propagation algorithm, why does the model performs well?  The window during the testing is the same as the shift windonw and the local information has been broken as in previous works.

4. L21-L47 in the Supplemental Material. Two algorithms mixed the dot production and matrix production.

[1] Uformer: A general u-shaped transformer for image restoration

[2] Swinir: Image restoration using swin transformer.

---

> ### Author Response · Authors · 2022-08-02
> **Response to Reviewer S7Qf**
>
> **Q1:** How does the testing step avoid the original expensive operations?
>
> **A1:** For testing, we design the layer expectation propagation (LEP) to approximate the expectation of the stochastic factors. The exact inference procedure is Eq. 4. Since the $(\xi^0_h, \xi^0_w,\ldots,\xi^{N-1}_h, \xi^{N-1}_w)$ has exponential combinations ($s^{2N}$, where $s$ is the window size), the exact calculation involves summation over items whose number grows exponentially with $N$ $(\Theta(s^{2N}))$. Earlier researches [1, 2] also introduced the stochastic factors into the models to solve their own chanllenges and proposed to approximate the exact expectation by layer-wise expectation. Inspired by these works, we further propose the LEP to approximate the exact inference.  Apparently, it involves $s^2$ items per layer. Hence, the complexity grows linearly with $N$ $(\Theta(Ns^2))$.
>
> **Q2:** Do the testing FLOPs increase compared with the training?
>
> **A2:** Due to the layer-wise expectation, the testing FLOPs are higher than those of training. The higher FLOPs derive from that all local windows are taken into account for a single forward propagation. But the increase in FLOPs is deserved because it brings: 1) consistent performance gains, 2) lower memory cost compared with the sliding window (please refer to A3), 3) translation invariance and locality. Moreover, we also found that LEP can offer more flexibility to the FLOPs-performance trade-off, as described in Sec. 4.6.
>
> **Q3:** The detailed comparison (FLOPs and memory cost) with the fixed and sliding window.
>
> **A3:** We provide a detailed analysis about time and space complexity of the stochastic window strategy. We compare our strategy with the fixed and sliding window for training and testing, respectively. The meaning of each symbol: $B$: batch size, $(H, W, C)$: feature size, $s$: local window size, $h$: number of heads. We also report FLOPs and memory cost on a single attention layer, in which $B=1, h=1, s=8, H=W=256, C=128.$ We use GTX 1080Ti GPU as the computing device. OOM means out of memory(>11G).
>
> Table 1: The complexity for training.
> |Strategy|Time Complexity|Space Complexity|FLOPs (G)|Memory(G)|Trans. Inva.|
> |:----|:----:|:---:|:---:|:---:|:---:|
> |Fix. Win.|$\Theta(BHWCs^2+BHWC^2)$|$\Theta(BHWC+BHWhs^2)$|5.4|1.3|False|
> |Slid. Win.|$\Theta(BHWCs^2+BHWC^2)$|$\Theta(BHWCs^2+BHWhs^2)$|5.4|OOM|True|
> |Sto. Win.|$\Theta(BHWCs^2+BHWC^2)$|$\Theta(BHWC+BHWhs^2)$|5.4|1.3|True|
>
> For training, our strategy is as efficient as the fixed window for both speed and memory cost, as shown in Tab. 1. But the fixed window loses the translation invariance and locality. Note that the space complexity of the sliding window is significantly higher than others, which causes OOM and prevents the network training.
>
> Table 2: The complexity for testing.
> |Strategy|Time Complexity|Space Complexity|FLOPs (G)|Memory(G)|Trans. Inva.|
> |:----|:----:|:---:|:---:|:---:|:---:|
> |Fix. Win.|$\Theta(BHWCs^2+BHWC^2)$|$\Theta(BHWC+BHWhs^2)$|5.4|1.2|False|
> |Slid. Win.|$\Theta(BHWCs^2+BHWC^2)$|$\Theta(BHWCs^2+BHWhs^2)$|5.4|OOM|True|
> |Sto. Win.|$\Theta(BHWCs^4+BHWC^2)$|$\Theta(BHWC+BHWhs^2)$|72.3|1.2|True|
>
> For testing, since it requires to exploit all the local windows for a single forward pass, the extra cost is required. As shown in Tab. 2, the sliding window spends more memory ($\times s^2$) while our strategy spends more computations $(\times s^2)$. We have added the complexity analysis to the SM (L54-63).
>
> **Q4:** Integrating with existing transformers (e.g., Uformer, swinIR).
>
> **A4:** To validate the effectiveness of our method, we adopt the proposed strategy on one of the most popular U-shaped network architecture in this work. In fact, Uformer also adopts the same architecture except its specific modulator design. We discard the modulator because it is not general enough and unnecessary to verify our strategy. We are conducting experiments to intergrate our strategy with more architectures, including SwinIR. However, due to the time issue, we have to defer related content to future version.
>
> **Q5:** why does the model performs well without LEP?
>
> **A5:** During training, the stochastic window can exploit all the local windows fairly while the fixed window relies attention on the specific local windows. This means that the fixed window suffers from the loss of local information and translation invariance, while the stochastic window can leverage both to achieve a better generalization ability. Without LEP, the testing procedure is equivalent to the fixed window. In this setting, the using of the stochastic window for training can  optimize the network better, resulting in improved performance.
>
> **Q6:**  The dot and matrix production are mixed in SM.
>
> **A6:** We have fixed this error and carefully polished the whole manuscript in the revision.
>
> [1] Dropout: a simple way to prevent neural networks from overfitting. JMLR, 2014.
>
> [2] Deep networks with stochastic depth. ECCV, 2016.

---

> > ### Author Response · Authors · 2022-08-08
> > **More discussions**
> >
> > Thank you for the careful evaluation of our paper. We hope that this answer your question, and we would be happy to provide additional details if needed.

---

> > > ### Comment · Reviewer_S7Qf · 2022-08-10
> > > **Feedbacks to the authors**
> > >
> > > Thanks for your patient reply. After reading the feedback from the authors, I understand how the method work.
> > >
> > > 1. The problem is interesting. But, as shown in Table. 2, the computational cost is extremely heavy (more than x10). While the proposed method costs less memory compared with the sliding window methods, it is not a good tradeoff between computational and memory costs, especially when we use the Cuda implementation.
> > >
> > > 2. Due to the limited time, the generalization of the method on other transformer-based backbones is still unclear.
> > >
> > > As a result, I am leaning towards rejecting this paper.

---

### Meta-Review · Area_Chair_rpuA · 2022-08-22

**Recommendation:** Accept
**Confidence:** Less certain

**Metareview:**

The paper proposes a new stochastic window strategy for image restoration. The stochastic window transformer layer is invariant to translations and is applied to the image degradation and mitigates loss of locality, hence making the approach potentially more robust.
The reviews of the papers were mixed, with strong pro- and contra. One point of contention was whether the layer made sense at all, that is whether the invariance assumption is correct for image restoration. Another point of criticism was the additional computational burden imposed by the stochastic window layer. On the other hand reviewers liked SOTA performance, the novelty of the presented ideas and the paper writing. The rebuttal phase addressed many issues raised by reviewers, including the equivariance vs. invariance issue.
In my opinion the paper is an interesting and elegant theoretical contribution. I agree that *invariance w.r.t. image degradation* is indeed the right concept. While the additional runtime overhead may prevent the paper's method from application in time intensive scenarios, I still think that the approach has enough practical and theoretical strong points to merit publication.

**Award:**

No

---

### Decision · Program_Chairs · 2022-09-14

Accept